# Data Augmentation of Contrastive Learning is Estimating Positive-incentive Noise

**Hongyuan Zhang** [1]   **Yanchen Xu** [2 3]   **Sida Huang** [4 3]   **Xuelong Li** [3]

## Abstract

Inspired by the idea of Positive-incentive Noise (*Pi-Noise* or *π-Noise*) that aims at learning the reliable noise beneficial to tasks, we scientifically investigate the connection between contrastive learning and $\pi$-noise in this paper. By converting the contrastive loss to an auxiliary Gaussian distribution to quantitatively measure the difficulty of the specific contrastive model under the information theory framework, we properly define the task entropy, the core concept of $\pi$-noise, of contrastive learning. It is further proved that the predefined data augmentation in the standard contrastive learning paradigm can be regarded as a kind of point estimation of $\pi$-noise. Inspired by the theoretical study, a framework that develops a $\pi$-noise generator to learn the beneficial noise (instead of estimation) as data augmentations for contrast is proposed. The designed framework can be applied to diverse types of data and is also completely compatible with the existing contrastive models. From the visualization, we surprisingly find that the proposed method successfully learns effective augmentations. Our code is available at `https://github.com/hyzhang98/PiNDA`.

## 1. Introduction

Contrastive learning (van den Oord et al., 2018; Wu et al., 2018; He et al., 2020; Chen et al., 2020), as a high-performance self-supervised framework, has been intensively investigated in recent years. The contrastive paradigm is even applied to large-scale vision-language models (Rad-

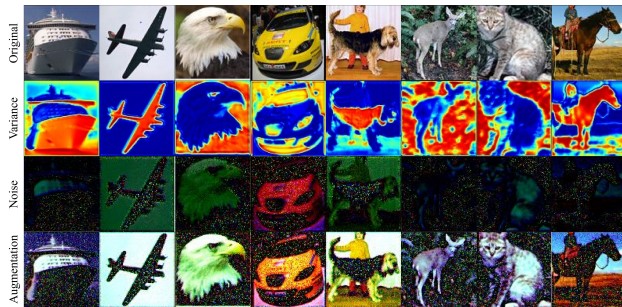

*Figure 1.* Visualization of the $\pi$-noise learned by SimCLR(Chen et al., 2020) with PiNDA on STL-10. We aim to learn the $\pi$-noise with a Gaussian distribution $\mathcal{N}(\boldsymbol{\mu}, \boldsymbol{\Sigma})$. We fix $\boldsymbol{\mu}$ as 0 and only learn the variance of Gaussian $\pi$-noise, which is visualized in the second row. The third row is a sampling noise and the fourth row contains the images added with $\pi$-noise. Compared with other visualizations, the noise is first normalized and then fed into the contrastive module. From the visualization, we find that the $\pi$-noise generator successfully learns effective visual augmentations (like style transfer) with only original images.

ford et al., 2021). One of the main limitations of the mainstream contrastive learning models is the dependency of data augmentation. In standard contrastive learning on vision data, the goal of data augmentation is to generate the positive pairs, which is the most crucial element of contrastive learning (Grill et al., 2020). The reliability of data augmentation decides the performance to a great extent (Chen et al., 2020). Inspired by human vision, there are plenty of high-quality augmentation schemes in computer vision, such as rotations, blurring, resizing, cropping, flipping, *etc*. Therefore, the performance of visual contrastive learning is the most remarkable. SimCLR (Chen et al., 2020), as a typical example, employs various strongly effective visual augmentations to construct positive pairs. CLIP (Radford et al., 2021), as a special example, seems to rely on no visual augmentation. However, as a vision-language model, the language view can be regarded as a cross-modal augmentation of the vision view.

On other types of data, how to generate stable augmentation views becomes one of the hottest research topics. Similar to CLIP, it is easy to extend the contrastive learning into multi-modal and multi-view data (Jiao et al., 2025; Pan & Kang, 2021). Take the graph data as an example: A pri-

---

[1]The University of Hong Kong [2]Fudan University [3]Institute of Artificial Intelligence (TeleAI), China Telecom [4]School of Artificial Intelligence, OPtics and ElectroNics (iOPEN), Northwestern Polytechnical University. Correspondence to: Xuelong Li <xuelong_li@ieee.org>.

*Proceedings of the $43^{rd}$ International Conference on Machine Learning*, Seoul, South Korea. PMLR 306, 2026. Copyright 2026 by the author(s).

mary limitation of graph contrastive learning (Hassani & Khasahmadi, 2020; Zhu et al., 2021) is the instability of random changing the graph structure as augmentations (*e.g.*, dropping edges and nodes, random walk). In the more general scenes where data can be only represented as vectors, some researchers (Cheung & Yeung, 2021; Verma et al., 2021; Yang et al., 2022; Hu et al., 2026) propose to directly utilize noise to generate the augmentations, which is also pointed out by SimCLR (Chen et al., 2020). For example, SimCL(Yang et al., 2022) simply uses random noise to generate the augmentation (Yang et al., 2022) for contrast while the performance is unstable and not satisfactory enough. DACL (Verma et al., 2021) proposes to apply the mixup noise with random ratio to generate positive pairs while MODALS (Cheung & Yeung, 2021) employs an automatic policy search algorithm to learn the mixup ratio for diverse augmentations. Overall, most existing studies of contrastive learning on non-vision data **only view the settings of noise as hyper-parameters** and fail to **learn** *what kind of noise will be beneficial*, which greatly limits the further promotion of contrastive learning. CLAE (Ho & Vasconcelos, 2020), using adversarial examples as positive pairs, can be regarded as an attempt to learn the profitable perturbation via a heuristic method since it simply uses the perturbation that maximizes the loss.

Inspired by the core idea of Positive-incentive Noise (*Pi-Noise* or *π-noise*) (Li, 2022; Zhang et al., 2023), we find that the framework of π-noise may offer a reliable scheme to learn stable data augmentations. Specifically speaking, the goal of π-noise is to find the beneficial noise by maximizing the mutual information between the task and noise, *i.e.*,

$$\max_{\mathcal{E}} \mathrm{MI}(\mathcal{T}, \mathcal{E}) = H(\mathcal{T}) - H(\mathcal{T}|\mathcal{E}), \tag{1}$$

where $\mathcal{T}$ denotes the current task, $\mathcal{E}$ is the noise set, and $\mathrm{MI}(\cdot, \cdot)$ is the mutual information of them. Note that $\mathcal{T}$ is an abstract notation of some random variable since the random variables differs a lot on diverse tasks. Formally, $\mathcal{E}$ subject to $\mathrm{MI}(\mathcal{T}, \mathcal{E}) > 0$ is named as π-noise, while the noise satisfying $\mathrm{MI}(\mathcal{T}, \mathcal{E}) = 0$ is pure noise (Li, 2022). In other words, compared with the random noise, π-noise will not disturb the task.

Motivated by the above intuition, we attempt to investigate **how to scientifically generate stable noisy augmentation views for general contrastive learning** in this paper. The contributions of this paper can be summarized as follows:

- At first, the task entropy of contrastive learning $H(\mathcal{T})$, the core of π-noise framework (Zhang et al., 2023), is defined by designing an auxiliary Gaussian distribution related to contrastive loss. The auxiliary distribution bridges the contrastive loss and information theory.

- With the proper definitions of $H(\mathcal{T})$, we prove that

the predefined data augmentations in the standard contrastive learning framework can be viewed as a point estimation of π-noise, where the prior augmentation is the point-estimated π-noise. From the theoretical analysis of the existing framework, we easily conclude that the point-estimated "π-noise" is easily designed but it is hard to extend to non-vision data, especially the data that can be only represented by vector.

- Motivated by the connection between π-noise and contrastive learning, we propose a π-noise generator to automatically learn the π-noise as the augmentation, instead of simply point-estimating the "π-noise" as the existing works, during training the representative module. Since the framework generates π-noise without depending on the specific form of data, the learned π-noise can be used as the augmentation of any type of data, which is not confined to vision data anymore.

Figure 1 shows the π-noise with Gaussian distributions learned by our proposed model. Note that the noise in Figure 1 is normalized to get a clearer visualization before it is fed to contrastive module. In other experiments, the noise is not normalized.

## 2. Background

### 2.1. Contrastive Learning

With the rise of self-supervised learning, contrastive learning (van den Oord et al., 2018; Wu et al., 2018; He et al., 2020; Chen et al., 2020; Grill et al., 2020) has been greatly studied. CPC (van den Oord et al., 2018) firstly develops the InfoNCE inspired by the NCE (Gutmann & Hyvärinen, 2010) for contrastive learning. Since the InfoNCE is applied to contrastive learning, plenty of works, such as Memory-Bank (Wu et al., 2018), Moco (He et al., 2020), and Sim-CLR (Chen et al., 2020), are proposed to greatly improve the performance. In particular, SimCLR proposes and verifies multiple crucial settings for contrastive models, *e.g.*, projection head, embedding normalization for InfoNCE. Some researchers (Robinson et al., 2021; Grill et al., 2020; Cao et al., 2022) focus on how to train the contrastive model more efficiently. For example, (Robinson et al., 2021) points out that more negative samples that are hard to distinguish can accelerate the training, while (Grill et al., 2020; Cao et al., 2022) propose to train the model with only positive samples and they achieve remarkable improvements of performance.

A main limitation of existing contrastive models is the dependency of high-quality reliable data augmentations. For example, SimCLR employs diverse visual augmentations (including rotation, cropping, resizing, blurring, flipping, *etc.*), and shows that data augmentations are extremely cru-

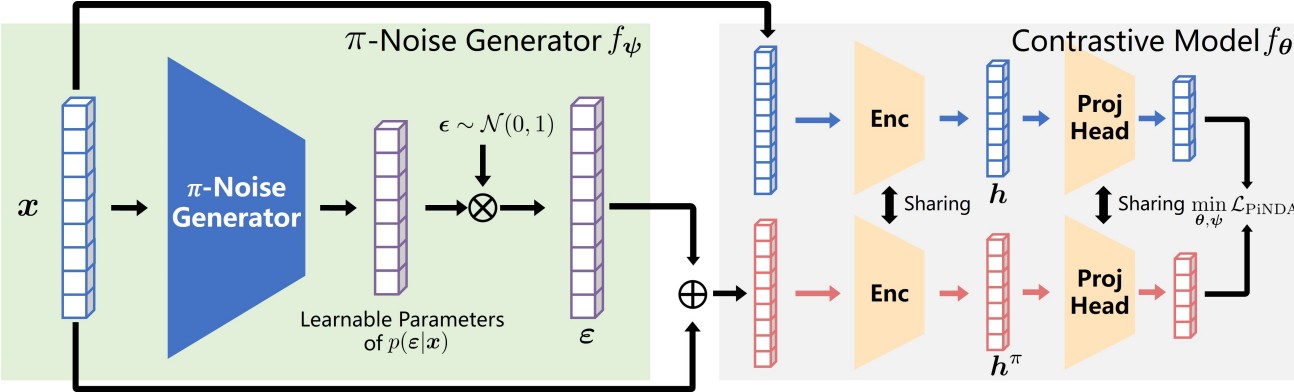

*Figure 2.* Framework of PiNDA: For simplicity, we assume that there is no other practicable augmentation. So the original sample and $\pi$-noise augmentation are used for contrast. If more data augmentations are available, the $\pi$-noise can be viewed as one of the augmentations and it will generate and backpropagate gradients to $\pi$-noise generator when the $\pi$-noise augmentation is sampled.

cial for the performance. When the data can be only represented as vectors, it is harder to obtain augmentations. SimCLR (Chen et al., 2020) finds that the utilization of noise can be a meaningful scheme. Some researchers (Yang et al., 2022; Ho & Vasconcelos, 2020; Verma et al., 2021; Cheung & Yeung, 2021; Wang et al., 2025) formally study how to apply noise to contrastive learning. SimCL (Yang et al., 2022) encodes the original features and then simply add normalized random noise for constructing positive pairs. DACL (Verma et al., 2021) mix up different samples with a random ratio, which is subject to uniform distribution, to generate positive pairs. MODALS (Cheung & Yeung, 2021) designs diverse predefined augmentations (including interpolation, extrapolation, Gaussian noise, and difference transform) and develops an automatic policy search strategy to determine the mixup ratio. CLAE (Ho & Vasconcelos, 2020) proposes to generate adversarial samples instead of random noise as an augmentation view. Overall, most works of how to extend contrastive learning into more general scenes simply view noise as predefined augmentations and fail to scientifically learn the stable and profitable noise in the continuous space.

### 2.2. Positive-incentive Noise

$\pi$-noise (Li, 2022) develops a framework based on information theory to formally claim that the noise may be not always harmful. The principle of maximum the mutual information between task and noise is proposed to learn the $\pi$-noise. The key definition in $\pi$-noise framework is the task entropy $H(\mathcal{T})$. For example, literature (Zhang et al., 2023; Huang et al., 2025) defines $p(y|\boldsymbol{x})$, the diversity of labels, as the task distribution to calculate $H(\mathcal{T})$ and then apply the variational inference to the optimization. Compared with it, this paper focuses on a more general and fundamental task, contrastive learning, which requires no supervision. The definition of task entropy in this paper is built on the loss

function, which is more general and scalable.

## 3. Proposed Method

In this section, we will first introduce the preliminary knowledge. Then the main theoretical derivations and conclusion of the relation between contrastive learning and $\pi$-noise is shown. Finally, the $\pi$-Noise Driven Data augmentations induced by the theoretical analysis will be elaborated. The whole framework is illustrated in Figure 2.

### 3.1. Preliminary of Contrastive Learning

Before the formal analysis, we first summarize the popular contrastive learning with InfoNCE (van den Oord et al., 2018) as follows

$$\min_{\boldsymbol{\theta}} -\frac{1}{|\mathcal{D}|} \sum_{\boldsymbol{x} \in \mathcal{D}} \log \frac{\ell_{\text{pos}}(\boldsymbol{x}; \boldsymbol{\theta})}{\ell_{\text{pos}}(\boldsymbol{x}; \boldsymbol{\theta}) + \ell_{\text{neg}}(\boldsymbol{x}; \boldsymbol{\theta})} = \mathcal{L}_{\text{InfoNCE}},$$
(2)

where $\ell_{\text{pos}}(\boldsymbol{x}; \boldsymbol{\theta})$ and $\ell_{\text{neg}}(\boldsymbol{x}; \boldsymbol{\theta})$ respectively denote the positive and negative loss of sample $\boldsymbol{x}$ parameterized by $\boldsymbol{\theta}$, $\mathcal{D}$ represents the dataset, and $|\mathcal{D}|$ is the dataset size. There are delicate differences of $\ell_{\text{pos}}$ and $\ell_{\text{neg}}$ among the mainstream contrastive paradigms (van den Oord et al., 2018; Wu et al., 2018; He et al., 2020; Chen et al., 2020; Radford et al., 2021). For instance, let $\boldsymbol{x}_+$, $\boldsymbol{x}_+'$ be the positive samples of $\boldsymbol{x}$, $f_{\boldsymbol{\theta}}$ be the learnable non-linear representative module, and $\tau$ be the temperature coefficient. The positive loss of SimCLR (Chen et al., 2020) is

$$\ell_{\text{pos}}^{\text{SimCLR}}(\boldsymbol{x}; \boldsymbol{\theta}) = \exp(\text{sim}(\boldsymbol{x}_+, \boldsymbol{x}_+')/\tau),$$
(3)

where $\text{sim}(\boldsymbol{u}, \boldsymbol{v}) = f_{\boldsymbol{\theta}}(\boldsymbol{u})^T f_{\boldsymbol{\theta}}(\boldsymbol{v})/(\|\boldsymbol{u}\| \cdot \|\boldsymbol{v}\|)$, while the MemoryBank (Wu et al., 2018) uses

$$\ell_{\text{pos}}^{\text{MB}}(\boldsymbol{x}; \boldsymbol{\theta}) = \exp(f_{\boldsymbol{\theta}}(\boldsymbol{x})^T f_{\boldsymbol{\theta}}(\boldsymbol{x}_+)/\tau)$$
(4)

as the positive loss. Similarly, the negative loss and choice of negative pairs are also technically different from each

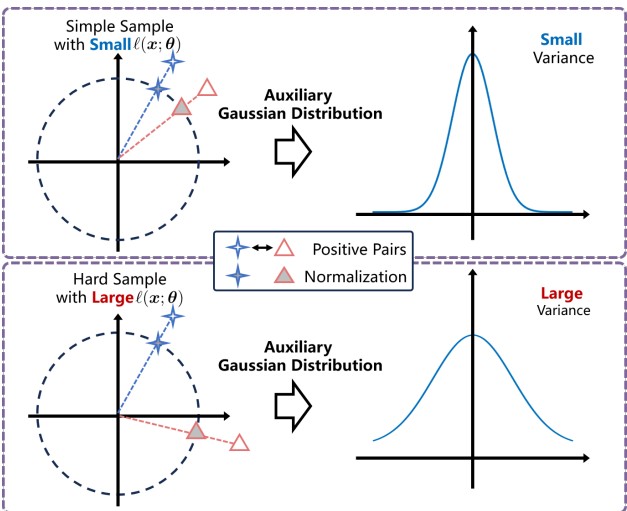

*Figure 3.* Illustration of the auxiliary Gaussian distribution. The smaller contrastive loss (*i.e.*, larger $\gamma_{\boldsymbol{\theta}^*}(\boldsymbol{x}, \boldsymbol{\varepsilon})$) is converted to a Gaussian distribution with smaller variance (*i.e.*, smaller entropy) and vice versa.

other.

To state the following analysis, we separate the contrastive loss and define the contrastive loss of each data point $\boldsymbol{x}$ as

$$\ell(\boldsymbol{x}; \boldsymbol{\theta}) = -\log \frac{\ell_{\text{pos}}(\boldsymbol{x}; \boldsymbol{\theta})}{\ell_{\text{pos}}(\boldsymbol{x}; \boldsymbol{\theta}) + \ell_{\text{neg}}(\boldsymbol{x}; \boldsymbol{\theta})}, \quad (5)$$

so that $\mathcal{L}_{\text{InfoNCE}} = 1/|\mathcal{D}| \cdot \sum_{\boldsymbol{x}} \ell(\boldsymbol{x}; \boldsymbol{\theta})$.

## 3.2. Formulate Task Entropy via Contrastive Loss

It seems intractable to precisely design the task entropy of contrastive learning with different contrastive models as the original definition. However, it seems much easier to measure the difficulty of contrastive learning on a **specific contrastive model** and a **given dataset**. Clearly, given a dataset, $\sum_{\boldsymbol{x}} \ell(\boldsymbol{x}; \boldsymbol{\theta}^*)$ is a straightforward quantity to reflect how difficult the contrastive learning task on the specific dataset is, where $\boldsymbol{\theta}^*$ is the optimal parameter. In other words, the smaller $\sum_{\boldsymbol{x}} \ell(\boldsymbol{x}; \boldsymbol{\theta}^*)$ means the easier contrastive task and vice versa. However, $\ell(\boldsymbol{x}; \boldsymbol{\theta}^*)$ is neither a random variable nor a probability quantity so that it can not be directly applied to Eq. (1). To bridge the framework of $\pi$-noise and the complexity metric $\ell(\boldsymbol{x}; \boldsymbol{\theta}^*)$, we introduce **an auxiliary random variable** $\alpha$ subject to $\alpha | \boldsymbol{x} \sim \mathcal{N}(0, \gamma_{\boldsymbol{\theta}^*}(\boldsymbol{x})^{-1})$. *i.e.*,

$$p(\alpha | \boldsymbol{x}) = \mathcal{N}(0, \gamma_{\boldsymbol{\theta}^*}(\boldsymbol{x})^{-1}), \quad (6)$$

where

$$\gamma_{\boldsymbol{\theta}^*}(\boldsymbol{x}) = \frac{\ell_{\text{pos}}(\boldsymbol{x}; \boldsymbol{\theta}^*)}{\ell_{\text{pos}}(\boldsymbol{x}; \boldsymbol{\theta}^*) + \ell_{\text{neg}}(\boldsymbol{x}; \boldsymbol{\theta}^*)} = \exp(-\ell(\boldsymbol{x}; \boldsymbol{\theta}^*)). \quad (7)$$

The information entropy of the above auxiliary distribution $p(\alpha | \boldsymbol{x})$ reflects the complexity of learning representation

of $\boldsymbol{x}$ via contrastive models. Accordingly, the task entropy of contrastive learning on a given distribution $p(\boldsymbol{x})$ can be formulated as

$$H(\mathcal{T}) = \mathbb{E}_{\boldsymbol{x} \sim p(\boldsymbol{x})} H(\mathcal{N}(0, \gamma_{\boldsymbol{\theta}^*}(\boldsymbol{x})^{-1})). \quad (8)$$

Note that $H(\mathcal{T})$ is lower-bounded by $H(\mathcal{N}(0, 1))$ due to that $\gamma_{\boldsymbol{\theta}^*}(\boldsymbol{x}) \in [0, 1]$. If $H(\mathcal{T})$ is required to reach the minimum of entropy, the definition of auxiliary variable can be $\kappa(\gamma_{\boldsymbol{\theta}^*}(\boldsymbol{x}))$, where $\kappa : [0, 1) \mapsto [0, \infty)$ is a a monotonically increasing function. However, the range of $H(\mathcal{T})$ does not affect the theoretical result and optimization. We thereby just use the simple formulation shown in Eq. (8).

Note that $\gamma_{\boldsymbol{\theta}^*}(\boldsymbol{x})$ is also related to the predefined data augmentations, while we omit it to simplify the notations and discussions. In the following subsections, we will show **the role of data augmentations under our framework**.

**Remark:** We omit the hyper-parameters (*e.g.*, temperature coefficient $\tau$ in Eq. (3) and (4)) above. It is easy and direct to extend the definitions to the loss with hyper-parameters. For example, the auxiliary distribution of the loss $\ell(\boldsymbol{x}; \boldsymbol{\theta}, \boldsymbol{\tau})$ can be directly defined as $\mathcal{N}(0, \gamma_{\boldsymbol{\theta}^*; \tau^*})$ where $\tau^*$ is the optimal hyper-parameter for the given model and dataset. However, to keep the notations uncluttered, the subscripts for hyper-parameters are omitted in the following part.

## 3.3. Connection of $\pi$-Noise and Contrastive Learning

According to the definition of the task entropy $H(\mathcal{T})$, the mutual information can be formulated as

$$\text{MI}(\mathcal{T}, \mathcal{E}) = \mathbb{E}_{\boldsymbol{x} \sim p(\boldsymbol{x})} \int p(\alpha, \boldsymbol{\varepsilon} | \boldsymbol{x}) \log \frac{p(\alpha, \boldsymbol{\varepsilon} | \boldsymbol{x})}{p(\alpha | \boldsymbol{x}) p(\boldsymbol{\varepsilon} | \boldsymbol{x})} d\alpha d\varepsilon$$

$$= \int p(\alpha, \boldsymbol{\varepsilon}, \boldsymbol{x}) \log \frac{p(\alpha, \boldsymbol{\varepsilon} | \boldsymbol{x})}{p(\alpha | \boldsymbol{x}) p(\boldsymbol{\varepsilon} | \boldsymbol{x})} d\alpha d\varepsilon d\boldsymbol{x}.$$

As shown in Eq. (1), the conditional entropy $H(\mathcal{T} | \mathcal{E})$ to minimize can be formulated as $H(\mathcal{T} | \mathcal{E}) =$

$$-\int p(\alpha | \boldsymbol{x}, \boldsymbol{\varepsilon}) p(\boldsymbol{\varepsilon} | \boldsymbol{x}) p(\boldsymbol{x}) \log p(\alpha | \boldsymbol{x}, \boldsymbol{\varepsilon}) d\boldsymbol{x} d\boldsymbol{\varepsilon} d\alpha. \quad (9)$$

Since the dataset $\mathcal{D}$ can be regarded as a sampling from $p(\boldsymbol{x})$, we can apply the Monte Carlo method to the above formulation and the conditional entropy can be estimated as $-H(\mathcal{T} | \mathcal{E}) \approx$

$$\frac{1}{n} \sum_{\boldsymbol{x}} \int p(\alpha | \boldsymbol{x}, \boldsymbol{\varepsilon}) p(\boldsymbol{\varepsilon} | \boldsymbol{x}) \log p(\alpha | \boldsymbol{x}, \boldsymbol{\varepsilon}) d\boldsymbol{\varepsilon} d\alpha. \quad (10)$$

On the right hand of the above formulation, there are two probabilities: $p(\alpha | \boldsymbol{x}, \boldsymbol{\varepsilon})$ and $p(\boldsymbol{\varepsilon} | \boldsymbol{x})$. $p(\boldsymbol{\varepsilon} | \boldsymbol{x})$ is the distribution (with some learnable parameters) of $\pi$-noise to learn. Therefore, it is a crucial step to model $p(\alpha | \boldsymbol{x}, \boldsymbol{\varepsilon})$. We first define $\ell_{\text{pos}}(\boldsymbol{x}, \boldsymbol{\varepsilon}; \boldsymbol{\theta}^*)$ and $\ell_{\text{neg}}(\boldsymbol{x}, \boldsymbol{\varepsilon}; \boldsymbol{\theta}^*)$ as the contrastive

loss with augmentation views **generated by** $\varepsilon$. Then, we can define the auxiliary distribution with $\varepsilon$ (shown in Section 3.2) as

$$p(\alpha|\boldsymbol{x}, \varepsilon) = \mathcal{N}(0, \gamma_{\boldsymbol{\theta}^*}(\boldsymbol{x}, \varepsilon)^{-1}), \quad (11)$$

where

$$\gamma_{\boldsymbol{\theta}^*}(\boldsymbol{x}, \varepsilon) = \frac{\ell_{\text{pos}}(\boldsymbol{x}, \varepsilon; \boldsymbol{\theta}^*)}{\ell_{\text{pos}}(\boldsymbol{x}, \varepsilon; \boldsymbol{\theta}^*) + \ell_{\text{neg}}(\boldsymbol{x}, \varepsilon; \boldsymbol{\theta}^*)}.$$

The intuition of the auxiliary distribution is illustrated in Figure 3.

To keep the simplicity of discussion, we assume that only one augmentation is used for contrast. It is not hard to verify that the following analysis still holds with multiple augmentations. If the data augmentation is viewed as a pre-defined special *noise*, then the augmentation is essentially equivalent to the following assumption

$$p(\varepsilon|\boldsymbol{x}) \to \delta_{\varepsilon_0}(\varepsilon), \quad (12)$$

where $\delta_{\varepsilon_0}(\varepsilon)$ is the Dirac delta function with translation $\varepsilon_0$, *i.e.*, $\delta_{\varepsilon_0}(\varepsilon)$ is not zero if and only if $\varepsilon = \varepsilon_0$. It is **the point estimation of** $p(\varepsilon|\boldsymbol{x})$, which is widely used in Maximum A Posteriori (MAP) (Bishop, 2007).

Owing to the point estimation, $-H(\mathcal{T}|\mathcal{E})$ can be further simplified as

$$-H(\mathcal{T}|\mathcal{E}) \approx \frac{1}{n} \sum_{\boldsymbol{x}}^{n} \int p(\alpha|\boldsymbol{x}, \varepsilon_0) \log p(\alpha|\boldsymbol{x}, \varepsilon_0) d\alpha = \mathcal{L}.$$

The probability density function (PDF) of $\mathcal{N}(0, \gamma_{\boldsymbol{\theta}^*}(\boldsymbol{x}, \varepsilon_0)^{-1})$ in $\mathcal{L}$ is formulated as

$$p(\alpha|\boldsymbol{x}, \varepsilon_0) = C\sqrt{\gamma_{\boldsymbol{\theta}^*}(\boldsymbol{x}, \varepsilon_0)} \exp(-\frac{\alpha^2}{2} \cdot \gamma_{\boldsymbol{\theta}^*}(\boldsymbol{x}, \varepsilon_0)),$$

where $C$ represents some constant term in Gaussian distribution. Substitute the PDF into $\mathcal{L}$, we have

$$\mathcal{L} = \frac{1}{n} \sum_{\boldsymbol{x}}^{n} \log C + \frac{1}{2} \log \gamma_{\boldsymbol{\theta}^*}(\boldsymbol{x}, \varepsilon_0) - \frac{1}{2}. \quad (13)$$

Due to the page limitation, more details can be found in Appendix A. Since the optimal $\boldsymbol{\theta}^*$ is not known and it is the learnable parameters of the contrastive model, it is a direct scheme to *simultaneously learn $\boldsymbol{\theta}$ during finding the $\pi$-noise $\mathcal{E}$*. Specifically, the formal optimization objective to find $\pi$-noise for contrastive learning is

$$\max_{\mathcal{E}, \boldsymbol{\theta}} \text{MI}(\mathcal{T}, \mathcal{E}). \quad (14)$$

Through the above derivations and estimations, the original goal to maximize the mutual information is converted to

$$\max_{\mathcal{E}, \boldsymbol{\theta}} \text{MI}(\mathcal{T}, \mathcal{E}) \Rightarrow \max_{\boldsymbol{\theta}} \mathcal{L} \Leftrightarrow \max_{\boldsymbol{\theta}} \frac{1}{n} \sum_{\boldsymbol{x}}^{n} \log \gamma_{\boldsymbol{\theta}}(\boldsymbol{x}, \varepsilon_0)$$

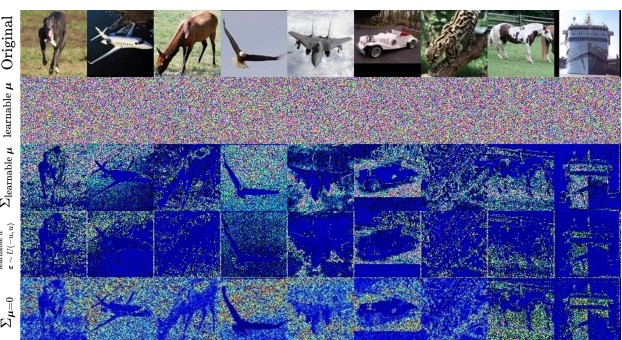

*Figure 4.* Visualization of different noise settings on SimCLR: In the second and third rows, we visualize the learnable $\boldsymbol{\mu}$ and $\boldsymbol{\Sigma}$ of Gaussian noise respectively. The fourth row shows the learned $\pi$-noise from the uniform distribution. The last row is the learned Gaussian noise with $\boldsymbol{\mu} = 0$.

$$\Leftrightarrow \min \frac{1}{n} \sum_{\boldsymbol{x}}^{n} -\log \frac{\ell_{\text{pos}}(\boldsymbol{x}, \varepsilon_0; \boldsymbol{\theta})}{\ell_{\text{pos}}(\boldsymbol{x}, \varepsilon_0; \boldsymbol{\theta}) + \ell_{\text{neg}}(\boldsymbol{x}, \varepsilon_0; \boldsymbol{\theta})}, \quad (15)$$

which is the same as the standard contrastive learning paradigm. In conclusion, the standard contrastive learning paradigm is equivalent to **optimize the contrastive learning module with point-estimated $\pi$-noise** where *the predefined data augmentation is the point estimation of $\pi$-noise*.

### 3.4. $\pi$-Noise Driven Data Augmentation

From the derivations presented in the previous subsection, there is a natural scheme to enhance the existing contrastive learning framework: automatically learn the $\pi$-noise, *i.e.*, Pi-Noise Driven Data Augmentation (*PiNDA*).

The advantages of learning $\pi$-noise as augmentations mainly includes: (1) Plenty of existing contrastive models (*e.g.*, SimCLR) also use the images with untrained noise as one of the predefined data augmentations and $\pi$-noise driven data augmentations offer a more stable scheme; (2) As the training of $\pi$-noise does not depend on the structure of data, it offers a reliable augmentation scheme for non-vision data. It should be emphasized that PiNDA is completely compatible with the existing contrastive models.

Formally, instead of applying the point-estimation of $\pi$-noise as shown in Eq. (12), we propose to directly learn $p_{\boldsymbol{\psi}}(\varepsilon|\boldsymbol{x})$ parameterized by $\boldsymbol{\psi}$. The negative conditional entropy estimated by the Monte Carlo method (given by Eq. (10)) can be rewritten as $-H(\mathcal{T}|\mathcal{E}) \approx$

$$\frac{1}{n} \sum_{\boldsymbol{x}} \int \mathbb{E}_{\varepsilon \sim p_{\boldsymbol{\psi}}(\varepsilon|\boldsymbol{x})} p(\alpha|\boldsymbol{x}, \varepsilon) \log p(\alpha|\boldsymbol{x}, \varepsilon) d\alpha. \quad (16)$$

The above expectation formulation can be also estimated by the Monte Carlo method. Furthermore, to backpropagate the gradient to $p_{\boldsymbol{\psi}}(\varepsilon|\boldsymbol{x})$, the reparameterization trick (Kingma & Welling, 2014) is applied. In other words, $p(\boldsymbol{\epsilon})d\boldsymbol{\epsilon} = p(\varepsilon|\boldsymbol{x})d\varepsilon$ where $\varepsilon = f_{\boldsymbol{\psi}}(\boldsymbol{x}, \boldsymbol{\epsilon})$, $f_{\boldsymbol{\psi}}$ is the $\pi$-noise generator,

**Algorithm 1** Pseudo code of compute the contrastive loss with PiNDA

---

**Input:** A sample $x$, batch size $m$, an augmentation $a(\cdot)$.
  Sample an auxiliary random noise $\epsilon \sim p(\epsilon)$.
  Generate $\pi$-noise: $\varepsilon = f_\psi(x, \epsilon)$.
  # $\pi$-noise augmentation
  Get a noisy augmentation $h^\pi = f_\theta(x + \varepsilon)$
  $h' = f_\theta(a(x))$.
  Compute $\mathcal{L}_{\text{PiNDA}}$ using $h^\pi$ and $h'$ as the condition $(x, \varepsilon)$ in Eq. (17).
  $\ell_{\text{PiNDA}}(x; \theta, \psi) = -\log \frac{\ell_{\text{pos}}(x, \varepsilon; \theta)}{\ell_{\text{pos}}(x, \varepsilon; \theta) + \ell_{\text{neg}}(x, \varepsilon; \theta)}$
**Output:** $\ell_{\text{PiNDA}}(x; \theta, \psi)$.

---

and $p(\epsilon)$ is a standard Gaussian distribution. Accordingly, the loss of PiNDA is formulated as

$$\mathcal{L}_{\text{PiNDA}} = -\frac{1}{n} \sum_x \mathbb{E}_{\epsilon \sim p(\epsilon)} \int p(\alpha|x, \varepsilon) \log p(\alpha|x, \varepsilon) d\alpha$$

$$= -\frac{1}{n} \sum_x \mathbb{E}_{\epsilon \sim p(\epsilon)} \log \frac{\ell_{\text{pos}}(x, \varepsilon; \theta)}{\ell_{\text{pos}}(x, \varepsilon; \theta) + \ell_{\text{neg}}(x, \varepsilon; \theta)}. \quad (17)$$

It should be pointed out that $\theta$ and $\psi$ represent the parameters of the contrastive model and $\pi$-noise generator, respectively. We summarize the whole procedure of a base contrastive model with PiNDA as the only augmentation in Algorithm 1. The pseudo code of a typical contrastive model with PiNDA and other predefined augmentations is summarized in Algorithm 2.

# 4. Experiments

In this section, we perform experiments to investigate *whether the proposed $\pi$-noise augmentation could promote the performance of the existing contrastive learning models*. As clarified in the previous sections, a primary merit of PiNDA is the generalization. In other words, PiNDA is not limited to vision data.

*Table 1.* Details of 4 non-vision datasets and 5 vision datasets.

| Dataset | Vision | Size | Classes |
|---|---|---|---|
| HAR | ✗ | 10,299 | 6 |
| Reuters | ✗ | 10,000 | 4 |
| Reuters-21578 | ✗ | 11,228 | 46 |
| Epsilon | ✗ | 500,299 | 2 |
| MSLR-WEB30K | ✗ | 10,558,186 | 5 |
| CIFAR-10 | ✓ | 60,000 | 10 |
| CIFAR-100 | ✓ | 60,000 | 100 |
| Fashion-MNIST | ✓ | 70,000 | 10 |
| STL-10 | ✓ | 113,000 | 10 |
| ImageNet | ✓ | 1,331,167 | 1000 |

**Algorithm 2** Pseudo code of PiNDA with other augmentations

---

**Input:** Dataset $\mathcal{D}$, batch size $m$, predefined augmentations $\mathcal{A}$.
  **if** $\mathcal{A} = \emptyset$ **then**
    # If no predefined augmentation is available, use the original view as another augmentation.
    $\mathcal{A} = \mathcal{A} \cup \{\text{Identity function}\}$.
  **end if**
  $\mathcal{A} = \mathcal{A} \cup \{\text{PiNDA}\}$.
  **for** A sampled batch $\{x_i\}_{i=1}^m$ **do**
    **for** each $x_i$ **do**
      Draw two augmentations $a \sim \mathcal{A}$ and $a' \sim \mathcal{A}$.
      **if** $\text{PiNDA} \in \{a, a'\}$ **then**
        Compute $\ell_{\text{PiNDA}}(x_i; \theta, \psi)$ by Algorithm 1.
        $\mathcal{L} = \mathcal{L} + \ell_{\text{PiNDA}}(x_i; \theta, \psi)$.
      **else**
        Compute $\ell(x_i; \theta)$ and $\mathcal{L} = \mathcal{L} + \ell(x_i; \theta)$.
      **end if**
    **end for**
    Update $\theta$ and $\psi$ to minimize $\mathcal{L}$.
  **end for**
**Output:** Contrastive model $f_\theta$ and $\pi$-noise generator $f_\psi$.

---

## 4.1. Datasets

In our experiments, there are 5 non-vision datasets used for validating the effectiveness of PiNDA on non-vision data: HAR (Reyes-Ortiz et al., 2012), Reuters and Reuters-21578 (Lewis et al., 2004), Epsilon (PASCAL, 2008), and MSLR-WEB30K (Qin & Liu, 2013). In addition, 5 vision datasets, including Fashion-MNIST (Xiao et al., 2017), CIFAR-10 (Torralba et al., 2008), CIFAR-100 (Torralba et al., 2008), STL-10 (Coates et al., 2011), and ImageNet (Russakovsky et al., 2015), are also used for evaluation. On the one hand, *the experiments on these vision datasets provide the better visualization of $\pi$-noise augmentation*. On the other hand, *the experiments also validates the effectiveness and compatibility of PiNDA to the existing contrastive models*. The details are summarized in Table 1. More details can be found in Appendix B.1.

## 4.2. Experimental Settings

**Baseline** For **non-vision datasets**, SimCL (Yang et al., 2022) and CLAE (Ho & Vasconcelos, 2020) serve as the competitors. *SimCL* adds the normalized noise to the learned representations to obtain augmentations, while *CLAE* (Ho & Vasconcelos, 2020) uses adversarial examples as an augmentation. To better *verify whether the idea of learning $\pi$-noise works*, a compared method that utilizes the untrained random Gaussian noise is also added as a competitor method, denoted by *Random Noise*. *SimCLR* (Chen et al., 2020),

*BYOL* (Grill et al., 2020), *SimSiam* (Chen & He, 2021), *MoCo* (He et al., 2020), and DINO (Caron et al., 2021) also serve as important baselines on **vision data**.

**Backbone** On non-vision datasets, the contrastive backbone is set to a 3-layer fully connected neural network since no efficient model can be applied. The dimension of each hidden layer is set to 1024 and the dimension of the final embedding is 256. On vision datasets, we use ResNet18 (He et al., 2016) as the backbone for contrastive representation learning for all the models by default. Moreover, we also use ResNet50 (He et al., 2016) as the backbone for some models to test the generalization of PiNDA.

*Table 2.* Test accuracy (%) averaged over 5 runs on 4 non-vision datasets evaluated by $k$NN and Softmax Regression (SR). All superscripts and subscripts represent the specific settings. The best results are highlighted in bold.

| Dataset | Method | $k$NN | SR |
|---|---|---|---|
| HAR | Random Noise | 77.76±0.61 | 77.62±0.27 |
| | SimCL | 61.12±3.21 | 63.92±0.16 |
| | PiNDA$_{\mu=0}$ | 77.14±0.94 | **86.20±0.40** |
| | PiNDA$_{\mu\neq0}$ | **78.09±0.75** | 81.33±0.60 |
| | CLAE | 85.71±0.18 | 90.80±0.66 |
| | PiNDA$_{CLAE}^{\mu=0}$ | **86.34±0.70** | **91.10±0.35** |
| | PiNDA$_{CLAE}^{\mu\neq0}$ | 85.89±0.01 | 90.52±0.59 |
| Reuters | Random Noise | 82.84±6.02 | 77.30±7.40 |
| | SimCL | 64.20±4.19 | 73.63±0.05 |
| | PiNDA$_{\mu=0}$ | 84.43±1.23 | **86.03±0.48** |
| | PiNDA$_{\mu\neq0}$ | **86.37±1.07** | 82.50±0.22 |
| | CLAE | 82.60±0.33 | 78.03±0.36 |
| | PiNDA$_{CLAE}^{\mu=0}$ | 82.37±1.07 | 79.03±0.66 |
| | PiNDA$_{CLAE}^{\mu\neq0}$ | **82.80±0.01** | **79.07±0.66** |
| Epsilon | Random Noise | 52.00±0.30 | 60.21±0.28 |
| | SimCL | 50.90±0.36 | 59.49±0.01 |
| | PiNDA$_{\mu=0}$ | **53.20±0.08** | **61.53±0.08** |
| | PiNDA$_{\mu\neq0}$ | 52.64±0.34 | 60.21±0.28 |
| | CLAE | 51.91±0.35 | 59.17±0.61 |
| | PiNDA$_{CLAE}^{\mu=0}$ | **52.31±0.29** | 59.38±0.32 |
| | PiNDA$_{CLAE}^{\mu\neq0}$ | 51.70±0.18 | **59.66±0.38** |
| MSLR -WEB30K | Random Noise | 57.88±1.49 | 33.78±6.31 |
| | SimCL | 64.21±0.42 | 47.13±0.78 |
| | PiNDA$_{\mu=0}$ | **69.62±0.02** | 49.55±0.11 |
| | PiNDA$_{\mu\neq0}$ | 69.60±0.06 | **50.95±1.43** |
| | CLAE | 67.99±0.78 | 51.54±0.10 |
| | PiNDA$_{CLAE}^{\mu=0}$ | 68.29±0.53 | **52.18±0.90** |
| | PiNDA$_{CLAE}^{\mu\neq0}$ | **68.66±1.12** | 51.79±0.77 |
| Reuters -21578 | Random Noise | 38.30±0.47 | 44.21±0.14 |
| | SimCL | 38.00±0.22 | 44.34±0.40 |
| | PiNDA$_{\mu=0}$ | 38.40±0.06 | **44.37±0.27** |
| | CLAE | 38.95±0.39 | 41.91±0.10 |
| | PiNDA$_{CLAE}^{\mu=0}$ | **39.18±0.26** | 42.07±0.50 |

**$\pi$-noise Setting** In this paper, we attempt to learn Gaussian $\pi$-noise, *i.e.*,

$$p(\boldsymbol{\varepsilon}|\boldsymbol{x}) = \mathcal{N}(\boldsymbol{\mu}, \boldsymbol{\Sigma}) \qquad (18)$$

where $\boldsymbol{\mu}$ and $\boldsymbol{\Sigma}$ represent the mean and variance respectively. To reduce the parameters to learn, we assume that $\boldsymbol{\Sigma}$ is a

diagonal matrix. In other words, $p(\boldsymbol{\varepsilon}|\boldsymbol{x})$ is an uncorrelated Gaussian distribution. On non-vision datasets, $\boldsymbol{\mu}$ and $\boldsymbol{\Sigma}$ are both encoded by a 3-layer fully connected neural network, namely $\pi$-noise generator. Formally speaking, $(\boldsymbol{\mu}, \boldsymbol{\Sigma}) = \mathrm{DNN}_{\boldsymbol{\psi}}(\boldsymbol{x})$ where $\boldsymbol{\psi}$ is the learnable parameters. The hidden layers of the $\pi$-noise generator are all 1024. On vision datasets, they are represented by a ResNet18 to better extract the visual information.

After sampling standard Gaussian variable $\boldsymbol{\epsilon}$, the $\pi$-noise variable is obtained by the reparameterization trick, *i.e.*,

$$\boldsymbol{\varepsilon} = \boldsymbol{\mu} + \boldsymbol{\epsilon} \odot \boldsymbol{\Sigma} \qquad (19)$$

To study different kinds of noise, we also test the noise with a uniform distribution. where $\odot$ is Hadamard product operator.

For a data point $\boldsymbol{x}$, the noise of each dimension $x_i$ obeys a independent uniform distribution, $p(\varepsilon|x_i) = U(-u, u)$ where $u$ is encoded by $x_i$. The experimental results are shown in Table 3 and Figure 4. The reparameterization trick is also used to retain the differentiability of noise sampling. Formally speaking, after sampling a noise $\epsilon$ from $U(0, 1)$, the $\pi$-noise is computed by $\varepsilon = (2\epsilon - 1) \cdot u$.

To study whether the mean vector makes sense, we added the ablation experiments with fixing $\boldsymbol{\mu} = 0$. As shown by our experiments, the improvements of learnable $\boldsymbol{\mu}$ may be instable but it will result in much more parameters and calculations. So we suggest to fix $\boldsymbol{\mu}$ as 0 when the computational resources are limited. The codes are implemented under PyTorch and the experiments are conducted on a single GPU. The source codes are available at https://github.com/hyzhang98/PiNDA.

### 4.3. Performance Analysis

#### 4.3.1. RESULTS ON NON-VISION DATASETS

The classification accuracy evaluated by $k$NN and Sotfmax Regression on HAR and Reuters is reported in Table 2. From the table, it is not hard to find that PiNDA achieves better performance than baselines. For example, PiNDA achieves about 5% $k$NN improvement and 9% Softmax Regression improvement on Reuters than Random Noise. In particular, on Reuters, it results in severe instability to use the simple random noise as a main augmentation, which is quantitatively shown by variances. SimCL achieves relatively worse results, which may be caused by the normalization of added noise. The normalization also results in instability. However, we find that *the normalized $\pi$-noise is more stable in Figure 1*. The comparison of PiNDA and PiNDA with $\boldsymbol{\mu} = 0$ implies that the constraint of learnable mean vector (or namely bias) is frequently beneficial. It may suffer from the over-fitting problem when $\boldsymbol{\mu}$ is not constrained. The experiments with CLAE verify that PiNDA is

*Table 3.* Test accuracy (%) averaged over 5 runs on CIFAR-10 and CIFAR-100. Without specific statements, epochs of base models are set as 1000 and the batch size (denoted by $m$) is 256. Note that the performance gaps are mainly caused by the limitation of GPU, which limits the batch size in our experiments. But it still shows that PiNDA always promotes the base models.

| Base Models | Setting | PiNDA Setting | Other Aug. | CIFAR-10 | | CIFAR-100 | |
|---|---|---|---|---|---|---|---|
| | | | | $k$NN | SR | $k$NN | SR |
| SimCLR | ResNet18, $m$=256 | — | ✗ | 28.89±0.46 | 36.92±0.36 | 5.93±0.37 | 9.58±0.24 |
| | | $\mathcal{N}(0, \Sigma)$ | ✗ | **32.48±0.48** | **37.57±0.98** | **6.46±0.33** | **10.13±0.17** |
| | | $\mathcal{N}(\boldsymbol{\mu}, \Sigma)$ | ✗ | 31.83±0.58 | 36.72±0.93 | 5.50±0.38 | 9.86±0.09 |
| | ResNet18, $m$=256 | — | ✓ | 86.05±0.21 | 91.98±0.71 | 37.56±0.80 | 70.43±0.27 |
| | | $\mathcal{N}(0, \Sigma)$ | ✓ | 86.60±0.39 | **92.25±0.07** | 37.62±0.64 | **70.64±0.44** |
| | | $\mathcal{N}(\boldsymbol{\mu}, \Sigma)$ | ✓ | **87.75± 0.60** | 90.50±0.47 | **38.86±0.11** | 68.96±0.11 |
| | ResNet50, $m$=64 | — | ✓ | 66.38±0.50 | 50.91±0.53 | **30.60±1.06** | 59.33±1.78 |
| | | $\mathcal{N}(0, \Sigma)$ | ✓ | **70.00±1.30** | **52.20±0.34** | 30.27±0.50 | **62.00±1.93** |
| | ResNet50, $m$=256 | — | ✓ | 74.43±0.52 | 84.81±0.91 | 43.69±1.00 | 57.67±2.20 |
| | | $\mathcal{N}(0, \Sigma)$ | ✓ | **75.00±0.12** | **86.48±0.97** | **45.09±0.99** | **59.97±1.33** |
| | | $U(-u, u)$ | ✓ | 74.52±0.29 | 85.49±0.54 | 44.16±0.49 | 59.37±3.31 |
| | ResNet50, $m$=512 | — | ✓ | 75.65±0.46 | 85.52±1.85 | 44.59±0.11 | 58.70±2.60 |
| | | $\mathcal{N}(0, \Sigma)$ | ✓ | **76.41±0.30** | **88.50±0.62** | **46.30±0.44** | **60.58±1.72** |
| MoCo | ResNet18 | — | ✓ | 71.31±0.50 | 72.22±0.21 | 41.50±0.38 | 35.13±0.31 |
| | | $\mathcal{N}(0, \Sigma)$ | ✓ | 71.71±0.19 | 72.56±0.28 | 41.05±0.30 | 35.70±0.17 |
| | | $\mathcal{N}(\boldsymbol{\mu}, \Sigma)$ | ✓ | **73.50±0.41** | **74.16±0.30** | **43.18±0.19** | **40.76±0.13** |
| BYOL | ResNet18, epoch=100 | — | ✓ | 71.08±0.78 | 78.55±0.30 | 34.05±1.23 | 49.04±0.65 |
| | | $\mathcal{N}(0, \Sigma)$ | ✓ | **71.18±0.27** | **78.65±0.29** | 34.54±1.08 | **49.42±0.69** |
| | | $\mathcal{N}(\boldsymbol{\mu}, \Sigma)$ | ✓ | 71.16±0.13 | 78.43±0.25 | **37.75±0.34** | 48.97±0.38 |
| | ResNet18, epoch=1000 | — | ✓ | 89.78±0.09 | 92.29±0.08 | 54.80±0.60 | 62.21±1.00 |
| | | $\mathcal{N}(0, \Sigma)$ | ✓ | 89.94±0.15 | **92.48±0.17** | **55.36±0.80** | **62.74±0.57** |
| | | $\mathcal{N}(\boldsymbol{\mu}, \Sigma)$ | ✓ | **89.97±0.11** | 92.20±0.11 | 54.35±0.33 | 62.30±0.26 |
| SimSiam | ResNet18, epoch=100 | — | ✓ | 70.51±0.47 | 74.11±0.43 | 37.91±0.21 | 54.02±0.46 |
| | | $\mathcal{N}(0, \Sigma)$ | ✓ | 70.52±0.33 | **74.40±0.33** | 38.07±0.14 | **54.40±0.09** |
| | | $\mathcal{N}(\boldsymbol{\mu}, \Sigma)$ | ✓ | **70.86±0.46** | 74.30±0.51 | **42.71±0.16** | 52.30±0.12 |
| | ResNet18, epoch=1000 | — | ✓ | 87.32±0.19 | 90.73±0.38 | 54.84±0.12 | 66.29±0.16 |
| | | $\mathcal{N}(0, \Sigma)$ | ✓ | **87.42±0.13** | **90.85±0.20** | **54.99±0.28** | **66.39±0.56** |
| | | $\mathcal{N}(\boldsymbol{\mu}, \Sigma)$ | ✓ | 87.29±0.15 | 90.77±0.43 | 54.79±0.17 | 66.31±0.19 |

*Table 4.* Test accuracy (%) averaged over 5 runs on Fashion-MNIST and STL-10. The batch size is set as 256. The subscripts and superscripts mean the base models and special settings.

| Dataset | Method | $k$NN | SR |
|---|---|---|---|
| Fashion | SimCL | 67.00±2.70 | 68.41±2.14 |
| | Random | 78.44±0.26 | 75.24±0.17 |
| | PiNDA$_{\boldsymbol{\mu}=0}$ | **79.35±0.93** | **79.09±1.16** |
| | PiNDA$_{\boldsymbol{\mu}\neq 0}$ | 79.08±0.96 | 78.95±0.87 |
| STL-10 | SimCL | 27.23±0.72 | 34.52±1.29 |
| | Random | 33.17±0.59 | 38.24±0.21 |
| | PiNDA$_{\boldsymbol{\mu}=0}$ | 33.27±0.62 | **39.90±0.96** |
| | PiNDA$_{\boldsymbol{\mu}\neq 0}$ | **33.47±0.78** | 39.44±0.86 |
| | BYOL | 79.07±0.13 | 83.00±0.24 |
| | PiNDA$_{\text{BYOL}}^{\boldsymbol{\mu}=0}$ | 79.24±0.21 | **83.81±0.27** |
| | PiNDA$_{\text{BYOL}}^{\boldsymbol{\mu}\neq 0}$ | **79.47±0.56** | 83.52±0.06 |
| | SimSiam | 59.62±0.54 | 84.79±0.22 |
| | PiNDA$_{\text{SimSiam}}^{\boldsymbol{\mu}=0}$ | 77.18±0.21 | **84.80±0.15** |
| | PiNDA$_{\text{SimSiam}}^{\boldsymbol{\mu}\neq 0}$ | **77.57±0.17** | 84.31±0.12 |
| | MoCo | 72.13±0.65 | 76.84±0.32 |
| | PiNDA$_{\text{MoCo}}^{\boldsymbol{\mu}=0}$ | 72.19±0.25 | 77.09±0.25 |
| | PiNDA$_{\text{MoCo}}^{\boldsymbol{\mu}\neq 0}$ | **73.45±0.31** | **77.66±0.25** |
| | SimCLR18 | 61.69±0.67 | 62.54±0.95 |
| | PiNDA$_{\text{SimCLR18}}$ | **62.31±0.25** | **63.68±0.19** |
| | PiNDA$_{\text{SimCLR18}}^{\text{uniform}}$ | 61.30±0.29 | 63.28±1.17 |
| | SimCLR18$_{m=512}$ | 64.92±1.14 | 65.98±1.78 |
| | PiNDA$_{\text{SimCLR18}}^{m=512}$ | **65.55±0.94** | **67.28±0.89** |
| | SimCLR50 | 64.43±0.25 | 75.03±0.19 |
| | PiNDA$_{\text{SimCLR50}}$ | **65.47±0.36** | **75.25±0.26** |

*Table 5.* Test accuracy (%) on ImageNet. The batch size is set as 256. Due to the limitation of computational resources, the base models are only trained within 100 epochs.

| Backbone | Base | PiNDA Setting | $k$NN | SR |
|---|---|---|---|---|
| ResNet18 | BYOL | — | 43.04 | 56.37 |
| | | $\mathcal{N}(0, \Sigma)$ | **43.25** | **56.57** |
| | SimSiam | — | 42.96 | 56.55 |
| | | $\mathcal{N}(0, \Sigma)$ | **43.11** | **56.78** |
| | MoCo | — | 50.83 | 64.25 |
| | | $\mathcal{N}(0, \Sigma)$ | **51.10** | **64.58** |
| | SimCLR | — | 52.05 | 64.18 |
| | | $\mathcal{N}(0, \Sigma)$ | **55.48** | **66.86** |
| Transformer | DINO | — | 55.44 | 60.65 |
| | | $\mathcal{N}(0, \Sigma)$ | **55.50** | **61.22** |

compatible with the existing contrastive methods and ***it can be an extra module for any popular contrastive models*** to further promote the performance and stabilize the model.

### 4.3.2. RESULTS ON VISION DATASETS

The classification accuracy evaluated by $k$NN and Softmax Regression on vision datasets is summarized in Tables 3, 4, and 5. Similar to results on non-vision datasets, PiNDA outperforms other baselines. For example, PiNDA increases Random Noise by almost 4 percent under $k$NN evaluation on CIFAR-10 and 4 percent under Softmax Regression

on Fashion-MNIST. PiNDA improves the SimCLR with ResNet50 on CIFAR-100 more than 2 percent under all settings. It also increases MoCo by more than 5 percent. More importantly, PiNDA with SimSiam causes almost 20 percent improvement on STL-10 under the $k$NN evaluation. Note that the reason why the baselines cannot achieve the similar results reported by their original papers is the limited batch size, which is greatly **limited by computational resources**. The batch size of SimCLR is only set as 64/256/512 while SimCLR is sensitive to batch size. SimCLR with different backbones are also shown in Table 3. Overall, from our extensive experiments, we found that *PiNDA is stable technique to augment the existing contrastive learning models under diverse settings.*

### 4.3.3. VISUALIZATION

The learned $\pi$-noise is visualized in Figures 1 and 4. As discussed above, we mainly report PiNDA with $\boldsymbol{\mu} = 0$. In Figure 1, the noise is first normalized and then fed to contrastive models. In Figure 4, the noise is not normalized. Meanwhile in Figure 4, we visualize the different settings of $\pi$-noise, including Gaussian noise and uniform noise ($\mathcal{N}(0, \boldsymbol{\Sigma}), \mathcal{N}(\boldsymbol{\mu}, \boldsymbol{\Sigma})$, and $U(-u, u)$). Due to the limitation of pages, more visualizations with different backbones and more discussions can be found in Appendix C.3.

### 4.3.4. DIFFERENT NOISE GENERATORS

To study the impact of different implementations of the noise generator, the experiments of SimCLR on two high-resolution vision datasets, STL-10 and ImageNet, are conducted and the experimental results are shown in Table 6. We employ ResNet18, ResNet50, and U-Net (Ronneberger et al., 2015) as the generator.

*Table 6.* Comparison with different implementation of noise generator on STL-10 and ImageNet.

| Dataset | Generator | $k$NN | SR | Time | FLOPs | Space |
|---|---|---|---|---|---|---|
| | — | 61.69 | 62.54 | 21s | 2.57e12 | 48G |
| STL-10 | ResNet18 | 62.31 | 63.68 | +21s | +2.60e12 | +20G |
| (ResNet18) | ResNet50 | 54.72 | 61.24 | +45s | +6.08e12 | +133G |
| | U-Net | **63.57** | **63.95** | +20s | +4.73e12 | +22G |
| | — | 52.56 | 63.72 | 46s | 6.05e12 | 151G |
| STL-10 | ResNet18 | 51.84 | 64.31 | +18s | +2.60e12 | +25G |
| (ResNet50) | ResNet50 | 54.59 | **65.85** | +25s | +6.00e12 | +113G |
| | U-Net | **55.92** | 63.48 | +19s | +4.80e12 | +33G |
| | — | 52.05 | 64.18 | 3min | 2.57e12 | 48G |
| ImageNet | ResNet18 | **55.48** | **66.86** | +4min | +2.60e12 | +20G |
| (ResNet18) | ResNet50 | 45.18 | 58.53 | +10min | +6.08e12 | +133G |
| | U-Net | 43.01 | 58.28 | +4min | +4.73e12 | +32G |
| | — | 52.03 | 65.85 | 9min | 6.05e12 | 151G |
| ImageNet | ResNet18 | 53.18 | 66.34 | +5min | +2.60e12 | +25G |
| (ResNet50) | ResNet50 | **57.41** | **67.70** | +10min | +6.00e12 | +113G |
| | U-Net | 56.04 | 67.40 | +4min | +4.80e12 | +33G |

### 4.3.5. EXPERIMENTS OF BATCH SIZE

As shown in SimCLR (Chen et al., 2020), larger batch size usually means better performance. However, due to the

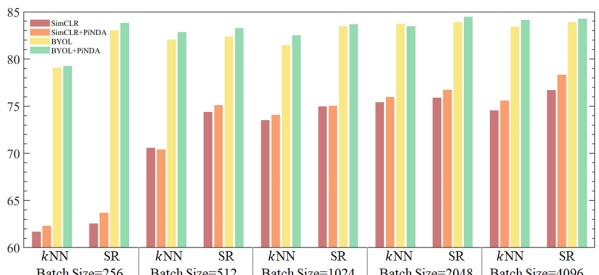

*Figure 5.* Test accuracy on STL10 with different batch sizes. Due to the computational limitation, the models are only trained within 100 epochs and the images are resized to $3 \times 32 \times 32$.

limitation of computational resources, we cannot reproduce many contrastive models with the large batch size as reported in the original papers. We try our best to conduct experiments with different batch sizes in Tables 2–5. Besides, experiments on STL-10 with several batch sizes are conducted to demonstrate the performance of PiNDA with different batch sizes. The results are shown in Figure 5. Note that due to the limitation of space, the images are resized to $3 \times 32 \times 32$ during the training.

## 5. Conclusion

In this paper, we design an auxiliary Gaussian distribution related to the contrastive loss to define the task entropy, the core of $\pi$-noise framework, of contrastive learning. With the definition of contrastive task entropy, we prove that the predefined data augmentations in the standard contrastive framework are equivalent to the point estimation of $\pi$-noise. Guided by the theoretical analysis, we propose to learn the $\pi$-noise instead of estimation. The learning of $\pi$-noise has two significant merits: (1) Since random noise is a common augmentation in existing contrastive learning, the proposed $\pi$-noise generator is completely compatible with the existing contrastive learning. Experiments show the existence and effectiveness of $\pi$-noise. (2) The generation of $\pi$-noise does not depend on the type of data and can be applied to any kind of data. The proposed method therefore helps extend contrastive learning to other fields. More surprisingly, we find that the developed $\pi$-noise generator successfully learns augmentations on vision datasets, which is similar to style transfer results. It indicates the potential of PiNDA and it deserves further investigation.

One may concern the additional burden of PiNDA. As shown in Table 11, PiNDA obtains better results than the original contrastive models under the similar computational expenses. In addition, as indicated by Table 6, the generator implemented the same architecture as the base model works well in most cases. In the future studies, a promising scheme is to share the parameters between the generator and base model, which can significantly reduce the extra expenses.

## Impact Statement

This paper presents work whose goal is to advance the field of machine learning. There are many potential societal consequences of our work, none of which we feel must be specifically highlighted here.

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

## A. Derivation of Eq. (13)

In this part, we show how to get Eq. (13). Substitute the PDF of $p(\alpha|\boldsymbol{x}, \boldsymbol{\varepsilon}_0)$ into $\mathcal{L}$, we have

$$
\begin{aligned}
\mathcal{L} &= \frac{1}{n} \sum_{\boldsymbol{x}}^{n} \int p(\alpha|\boldsymbol{x}, \boldsymbol{\varepsilon}_0) \cdot (\log C + \frac{1}{2} \log \gamma_{\boldsymbol{\theta}^*}(\boldsymbol{x}, \boldsymbol{\varepsilon}_0) - \frac{\alpha^2}{2} \cdot \gamma_{\boldsymbol{\theta}^*}(\boldsymbol{x}, \boldsymbol{\varepsilon}_0)) d\alpha \\
&= \frac{1}{n} \sum_{\boldsymbol{x}}^{n} \log C + \frac{1}{2} \log \gamma_{\boldsymbol{\theta}^*}(\boldsymbol{x}, \boldsymbol{\varepsilon}_0) - \frac{\gamma_{\boldsymbol{\theta}^*}(\boldsymbol{x}, \boldsymbol{\varepsilon}_0)}{2} \int \alpha^2 p(\alpha|\boldsymbol{x}, \boldsymbol{\varepsilon}_0) d\alpha \\
&= \frac{1}{n} \sum_{\boldsymbol{x}}^{n} \log C + \frac{1}{2} \log \gamma_{\boldsymbol{\theta}^*}(\boldsymbol{x}, \boldsymbol{\varepsilon}_0) - \frac{1}{2}.
\end{aligned}
\tag{20}
$$

The final step is derived from $\int \alpha^2 p(\alpha|\boldsymbol{x}, \boldsymbol{\varepsilon}_0) d\alpha = \mathbb{E}(\alpha - \mathbb{E}\alpha)^2 = \text{variance}(\alpha) = \gamma_{\boldsymbol{\theta}^*}(\boldsymbol{x}, \boldsymbol{\varepsilon}_0)^{-1}$.

## B. More Details of Experimental Settings

### B.1. Datasets

In our experiments, there are 4 non-vision datasets used for validating the effectiveness of PiNDA on non-vision data: HAR (Reyes-Ortiz et al., 2012) is collected by sensors of smartphones; Reuters (Lewis et al., 2004) is a subset of the original dataset and the preprocessing is the same with (Wu et al., 2022); epsilon (PASCAL, 2008) is a dataset used for PASCAL Challenge 2008; MSLR-WEB30K (Qin & Liu, 2013), consists of feature vectors extracted from query-url pairs along with relevance judgment labels, is released by Microsoft. Besides, 5 vision datasets, including Fashion-MNIST (Xiao et al., 2017), CIFAR-10 (Torralba et al., 2008), CIFAR-100 (Torralba et al., 2008), STL-10 (Coates et al., 2011), and ImageNet (Russakovsky et al., 2015), are also used for evaluation. On the one hand, *the experiments on these vision datasets provide the better visualization of $\pi$-noise augmentation*. On the other hand, *the experiments also validates the effectiveness and compatibility of PiNDA to the existing contrastive models*. The details are summarized in Table 1. Without additional clarifications, the images of STL-10 are not resized and the images of ImageNet are resized to $3 \times 96 \times 96$.

### B.2. Baseline

*SimCL* (Yang et al., 2022) proposes to add the normalized noise to the learned representations to obtain augmentations and feed the noisy augmentations into the projection head to compute the contrastive loss. Therefore, it is an important baseline. To better *verify whether the idea of learning $\pi$-noise works*, a compared method that utilizes the untrained random Gaussian noise is also added as a competitor method, denoted by *Random Noise*. As the value range of original features of various datasets differs a lot, we first rescale the input feature and then add the standard Gaussian noise to the features. *PiNDA* denotes the contrastive model with only a $\pi$-noise augmentation. Since *CLAE* (Ho & Vasconcelos, 2020) uses adversarial examples as an augmentation and is not limited to non-vision datasets, it is another important baseline. *SimCLR* (Chen et al., 2020), *BYOL* (Grill et al., 2020), *SimSiam* (Chen & He, 2021), and *MoCo* (He et al., 2020) also serve as important baselines on vision data.

We modify the baselines by using $\pi$-noise as an additional augmentation for contrast, which are marked by subscripts. They are also conducted to verify whether PiNDA is compatible with other contrastive models. For all baselines, the backbone used for extracting contrastive representations is the same by default. The backbone networks on non-vision and vision datasets are 3-layer DNN and ResNet18, respectively.

Following SimCLR, we also use the projection head, which consists of a 256-dimension fully connected layer with ReLU activation and a 128-dimension linear layer, to minimize the contrastive loss and use the input of the projection head as the learned representation.

### B.3. Evaluation Settings

The extracted contrastive embeddings are testified by two simple machine learning models, softmax regression (a neural network composed of a linear layer and a softmax layer) and $k$-nearest neighbors ($k$NN). For the softmax regression, the epoch is set as 50 and the batch size is 256. and the learning rate is set as 0.001. For $k$NN, $k$ is set as 5. All experiments share the same settings. The split of the training set and test set is following the standard split of the datasets.

### B.4. Contrastive Loss

On non-vision datasets, the implementation details of the contrastive loss are the same as SimCLR. On vision datasets, the contrastive loss consists of the original loss according to the base model used and an additional penalty term, which is the multiplicative inverse of the average norm of the $\pi$-noises, ensuring that the norm of the $\pi$-noise will not be too small. The loss is optimized by the Adam algorithm (Kingma & Ba, 2015) with a learning rate of $10^{-3}$. The temperature coefficient $\tau$ in the contrastive loss is set to 0.1 on all datasets.

## C. More Experimental Results

In this section, we show more experimental results, visualizations, and experimental analysis. In Section C.2, the experiments with SAM (Kirillov et al., 2023) are reported, which serves as another baseline. In Section C.5, the additional costs of PiNDA with different settings are reported.

### C.1. More Analysis and Conclusions on Vision Datasets

From the experiments on vision datasets, it is clearly shown that $\mu = 0$ **may be a cost-efficient setting** in practice when $\pi$-noise is assume as Gaussian noise. It seems that the learnable $\mu$ will not bring further improvements in all cases but results in much more learnable parameters and computations compared with the setting $\mu = 0$. Therefore, we recommend to set $\mu = 0$ when the computational resources are limited.

### C.2. Comparison with SAM

From the visualization shown by Figures 1, 4, 6, and 7, we found that the noise generator can detect the irrelevant background and main objects. A direct idea is to compare PiNDA with Segment Anything Model (SAM) (Kirillov et al., 2023).

*Table 7.* Test accuracy (%) averaged over 5 runs on CIFAR-10, CIFAR-100, and STL-10. Epochs of base models are set as 1000.

| Backbone | Other Augmentations | PiNDA Setting | CIFAR-10 | | CIFAR-100 | | STL-10 | |
|---|---|---|---|---|---|---|---|---|
| | | | $k$NN | SR | $k$NN | SR | $k$NN | SR |
| ResNet18, $m$=256 | Default | — | 86.05±0.21 | 91.98±0.71 | 37.56±0.80 | 70.43±0.27 | 61.69±0.67 | 62.54±0.95 |
| | SAM | — | 66.61± 0.43 | 90.31±0.39 | 35.06±0.64 | 68.90±0.26 | 52.68±1.53 | 62.78±1.15 |
| | PiNDA | $\mathcal{N}(0, \Sigma)$ | **86.60±0.39** | **92.25±0.07** | **37.62±0.64** | **70.64±0.44** | **62.31±0.25** | **63.68±0.19** |
| ResNet50, $m$=256 | Default | — | 74.43±0.52 | 84.81±0.91 | 43.69±1.00 | 57.67±2.20 | 64.43±0.25 | 75.03±0.19 |
| | SAM | — | 68.35±0.56 | 84.36±2.09 | 37.81±0.80 | 57.56±2.13 | **72.83±3.58** | 75.04±2.70 |
| | PiNDA | $\mathcal{N}(0, \Sigma)$ | **75.00±0.12** | **86.48±0.97** | **45.09±0.99** | **59.97±1.33** | 65.47±0.36 | **75.25±0.26** |

*Table 8.* Test accuracy (%) on ImageNet. The batch size is set as 256. The base models are only trained within 100 epochs.

| Backbone | Base | Aug | Setting | $k$NN | SR |
|---|---|---|---|---|---|
| | | Default | — | 52.05 | 64.18 |
| ResNet18 | SimCLR | SAM | — | 55.41 | **66.95** |
| | | PiNDA | $\mathcal{N}(0, \Sigma)$ | **55.48** | 66.86 |

We use SAM to segment the background and then add noise to the background with the highest probability. All processing with SAM is conducted before the training as a pre-processing step. The experiments are summarized in Table 7 and Table 8. From the two additional tables, we found that in most cases, PiNDA is more effective than SAM. It should be emphasized that **the pre-processing of SAM on ImageNet consumes more than 19 hours (about 6 times of training time)**, though it outperforms PiNDA on ImageNet. In contrast, the additional time required by PiNDA is only the half of training time.

### C.3. More Visualization and Discussions

In Figure 6, we show the unnormalized Gaussian $\pi$-noise with SimCLR on STL10. In Figure 7, we visualize the noise learned by a different contrastive model, BYOL.

Surprisingly, after sampling a $\pi$-noise and adding it to the original images, we obtain some remarkable augmentations. As shown in the figures, the learned $\pi$-noise indicates great effect toward the background of the image while mere effect toward the target, which leads to the bigger chance for the base model to be relevant to the target itself when extracting

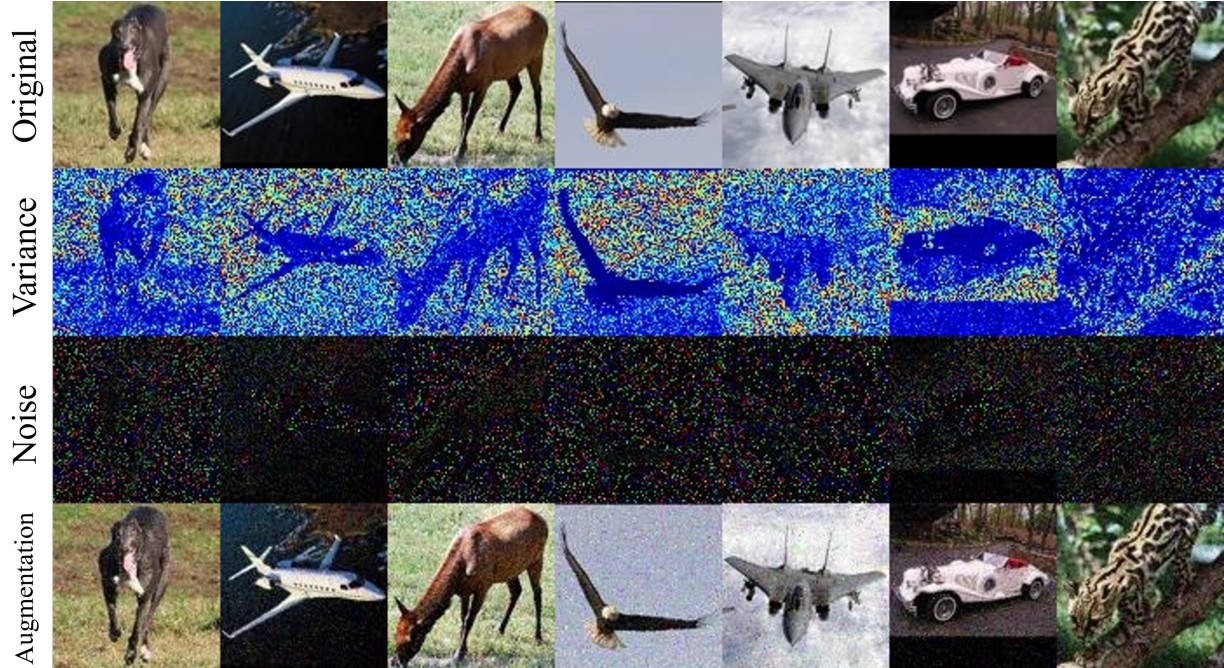

*Figure 6.* More visualization of the $\pi$-noise by PiNDA trained with SimCLR on STL-10. We fix $\boldsymbol{\mu}$ as 0 and only learn the variance of Gaussian $\pi$-noise, which is visualized in the second row. The third row is a sampling result from the learned $\pi$-noise distribution. Compared with Figure 1, the noise is not normalized.

features. For example, in Figure 1, the rightmost image (horse) is augmented by changing the sky (bluer) and ground (from grassland to desert), which is extremely similar to the style transfer result. Nevertheless, with only the original image inputted into the networks, the $\pi$-noise generator extracts the effective visual information and produce strong augmentations, which significantly validates the idea. The visualization also indicates the great potential and underlying applications of the proposed $\pi$-noise generators. In Figure 6, we found that the noise generator can still effectively learn the outline of objects.

More surprisingly, in Figure 4, we found that the learnable $\boldsymbol{\mu}$ contains no outline information and only $\boldsymbol{\Sigma}$ contains the useful information. It partially explains why learnable $\boldsymbol{\mu}$ will not alway improve the performance, which is shown in Tables 3 and 4. We also found that the uniform $\pi$-noise generator can also learn the outline but the visualization is not good as Gaussian $\pi$-noise.

From Figure 7, we find that PiNDA with other contrastive models (even without negative samples) can also learn the outline of objects in an unsupervised way.

Furthermore, the learned $\pi$-noise demonstrates the ability to differ the background from target that has similar color. For example, in the rightmost image (cat) of Figure 6, the variance shows the trend of the noise to influence the branch much more than to influence the cat. The same occasion also occurs in the rightmost two images of Figure 7. In the images, although the main body of the monkey and the car have similar color to the surrounding, the visualization of variance indicates that PiNDA successfully differs them from the background.

**C.4. More Analysis about Different Noise Generators (Table 6)**

To study the impact of different implementations of the noise generator, the experiments of SimCLR on two high-resolution vision datasets, STL-10 and ImageNet, are conducted and the experimental results are shown in Table 6. We employ ResNet18, ResNet50, and U-Net (Ronneberger et al., 2015) as the generator.

From Table 6, we can conclude that ResNet-18 is the most appropriate models from the aspects of both performance and costs. Although ResNet-50 outperforms ResNet-18 in some cases, it requires too much additional overhead but the improvement is relatively limited. U-Net is harder to train on ImageNet and results in more computational overhead compared with ResNet-18.

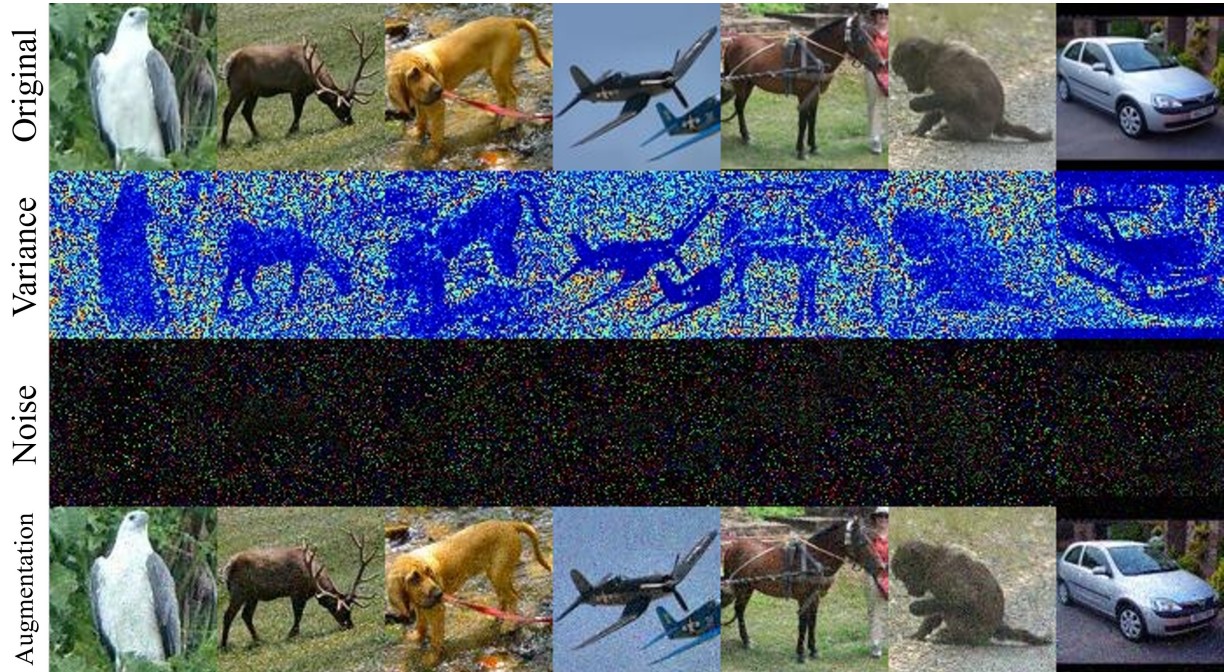

*Figure 7.* Visualization of the $\pi$-noise by PiNDA trained with BYOL on STL-10. Similarly, we fix $\mu$ as 0 and only learn the variance of Gaussian $\pi$-noise. We find that PiNDA can effectively learn the outline with different contrastive framework.

*Table 9.* Additional computational costs of SimCLR. Time: Training time per epoch. Memory: GPU memory. Batch size: 256.

| Generator | CIFAR-10 | | | CIFAR-100 | | | STL-10 | | | ImageNet | | |
|---|---|---|---|---|---|---|---|---|---|---|---|---|
| | Time | FLOPs | Memory | Time | FLOPs | Memory | Time | FLOPs | Memory | Time | FLOPs | Memory |
| Baseline | 39s | 1.32e9 | 15.85G | 39s | 1.32e9 | 15.85G | 21s | 2.57e12 | 48.20G | 3min | 2.57e12 | 48.20G |
| DNN3 | +5s | +1.26e7 | +0.23G | +5s | +1.26e7 | +0.23G | +11s | +5.64e9 | +11.47G | +2min | +5.64e9 | +11.47G |
| ResNet18 | +11s | +5.65e8 | +2.26G | +11s | +5.65e8 | +2.26G | +21s | +2.60e12 | +20.15G | +4min | +2.60e12 | +20.15G |
| ResNet50 | +36s | +1.32e9 | +12.55G | +36s | +1.32e9 | +12.55G | +45s | +6.08e12 | +133.03G | +10min | +6.08e12 | +133.03G |

## C.5. Additional Cost

Since PiNDA works as an additional module of contrastive models, the extra computational costs of training noise generator are important in practice. We report the additional costs to SimCLR with different implementations of noise generators on 4 vision datasets in Table 9. The additional costs to a DNN3 on 4 non-vision datasets are also reported in Table 10. The costs are evaluated by 3 different metrics: training time, floating point of operations (FLOPs), and required GPU memory. Note that the noise generator is implemented by ResNet18 in all other experiments though we test the computational costs of ResNet-50 as well. If we use ResNet-50 as the generator, the out-of-memory (OOM) exception will occur in many cases.

*Table 10.* Additional computational costs on non-vision datasets.

| Dataset | Generator | Time | FLOPs | Memory |
|---|---|---|---|---|
| HAR | Baseline | 0.23s | 4.99e8 | 900M |
| | PiNDA | +0.08s | +1.25e9 | +84M |
| Reuters | Baseline | 0.43s | 8.76e8 | 1044M |
| | PiNDA | +0.05s | +2.38e9 | +146M |
| Epsilon | Baseline | 17.23s | 1.73e9 | 7.04G |
| | PiNDA | +0.90s | +2.38e9 | +1.12G |
| MSLR -WEB30K | Baseline | 136.56s | 3.88e8 | 6.72G |
| | PiNDA | +55.45s | +9.12e8 | +0.32G |

C.5.1. PERFORMANCE COMPARISONS WITH SIMILAR BURDEN

We conduct further experiments to show the effectiveness of PiNDA with low computational cost. The results are shown in Table 11. It is easy to find that PiNDA can achieve better results compared with the original models with larger batch size,

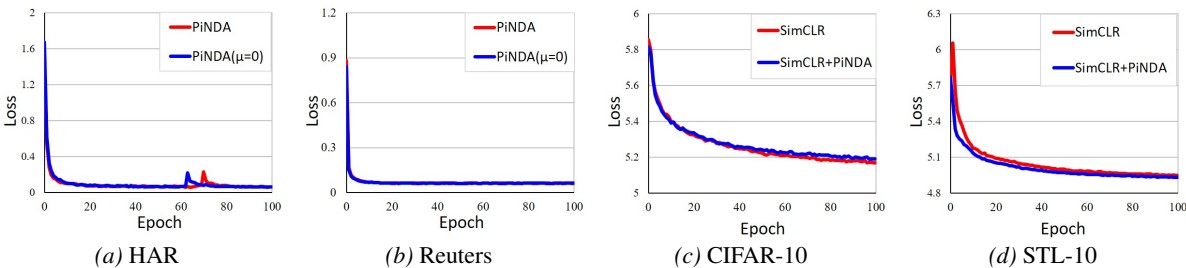

|  (a) HAR  |  (b) Reuters  |  (c) CIFAR-10  |  (d) STL-10  |

*Figure 8.* Loss curves on two non-vision datasets and two vision datasets. From the figures, we find that the loss of PiNDA decreases rapidly and converges fast.

*Table 11.* Comparison of accuracy of SimCLR and SimCLR with PiNDA under similar memory burden condition.

|  |  | CIFAR-10 | | |  | CIFAR-100 | | |
|---|---|---|---|---|---|---|---|---|
|  | Batch Size | Memory | $k$NN | SR | Batch Size | Memory | $k$NN | SR |
| SimCLR | 256 | 15.85G | 74.43±0.52 | 84.81±0.91 | 256 | 15.70G | 43.59±1.00 | 57.67±2.20 |
|  | 300 | 19.06G | 72.44±0.23 | 85.76±0.55 | 300 | 19.07G | 43.83±0.83 | 58.82±1.60 |
| PiNDA$_{SimCLR}$ | 256 | 18.11G | **75.00±0.12** | **86.48±0.97** | 256 | 18.10G | **45.09±0.99** | **59.97±1.33** |

when they have the same cost. It indicates that **the improvement brought by PiNDA is more effective and direct than the benefits from larger batch**.

## C.6. Convergence

To study the convergence of PiNDA, the loss curves on HAR, Reuters, CIFAR-10, and STL-10 are shown in Figure 8. From the figure, it is easy to find that the loss of PiNDA decreases rapidly. On STL-10, PiNDA helps the base SimCLR converge more rapidly. Although the loss is not a fair metric to reflect the performance, it empirically verifies the convergence of PiNDA.

