# OpenReview forum: "Data Augmentation of Contrastive Learning is Estimating Positive-incentive Noise"
_ICML.cc/2026/Conference — ICML 2026 regular_

### Official Review · Reviewer_gLQb · 2026-03-12

**Soundness:** 3
**Presentation:** 3
**Significance:** 2
**Originality:** 3
**Overall Recommendation:** 4
**Confidence:** 3

**Summary:**

Authors present a positive-incentive noise driven data augmentation (PiNDA) method designed to improve self-supervised contrastive learning.  The paper scientifically investigates how data augmentation in contrastive learning can be viewed as an estimation of $\pi$-noise. By converting contrastive loss into an auxiliary Gaussian distribution, the authors define the task entropy of contrastive learning within an information theory framework. The research addresses a major limitation in current contrastive learning: the heavy dependency on manually designed, high-quality data augmentations that are difficult to define for non-vision data. Extensive experiments show that PiNDA is completely compatible with existing models like SimCLR and BYOL, consistently enhancing their performance across diverse data types. Furthermore, visualizations reveal that the learned noise can effectively perform complex tasks such as unsupervised object segmentation and style transfer.

**Compliance With Llm Reviewing Policy:**

Affirmed.

**Key Questions For Authors:**

Authors are requested to explain

1.	Whether other non-Gaussian assumption for auxiliary variable significantly alter the proof that standard augmentation is a point estimation of pi-noise?

2.	Does the method provide a way to decode what a learned pi-noise distribution says about the optimal manual augmentations for a new, unknown dataset?

3.	The loss function includes a specific penalty term to prevent the generated noise from becoming too small. How sensitive is the final performance to the weight of this penalty, and is there a theoretical optimal noise magnitude that the generator should aim for?

4.	Are there any architectural optimizations, such as weight-sharing between the generator and the encoder or using lightweight distillation, that could allow for higher-capacity noise generation without the heavy memory burden?

5.	The results show that fixing the mean vector at zero is more stable than making it learnable. Beyond the increased parameter count, what is the underlying cause of the instability or overfitting observed when the mean is learnable?

6.	For non-vision data like the HAR or Reuters datasets, the noise is learned in a continuous vector space. How does the method handle datasets with discrete or categorical features, where adding Gaussian noise might result in invalid data points?

7.	Whether the learned noise patterns are consistent across different contrastive backbones for the same dataset, or does each model (e.g., SimCLR vs. BYOL) learn a preferred type of noise?

**Limitations:**

Yes, the authors provide a thorough discussion of the technical and computational limitations of the proposed method, such as computational overhead, memory constraints, parameter instability, optimization sensitivity, and hardware dependency but they do not detail specific potential negative societal impacts.

**Strengths And Weaknesses:**

Strengths

•	The method establishes a strong theoretical foundation by bridging contrastive learning and information theory through the formal definition of task entropy. A major advantage is its universal applicability; because it generates noise in a continuous space, it provides a reliable augmentation scheme for non-vision data that lacks intuitive visual heuristics like cropping or rotation.

•	The method is completely compatible with existing models such as SimCLR, BYOL, and MoCo, allowing it to function as an additional module that consistently improves performance.

•	The method has shown significant gains, including a nearly 20 percent performance increase for SimSiam on the STL-10 dataset. It is also more effective at improving performance than simply increasing batch sizes under similar memory constraints.

•	The method can learn to distinguish objects from backgrounds in an unsupervised manner, enabling it to perform tasks like style transfer and segmentation without manual labels.

Weaknesses

•	The primary weakness is the increased computational overhead, as the noise generator requires its own parameters and operations, leading to higher training times, FLOPs, and GPU memory usage.

•	The method can be sensitive to its configuration; using a learnable mean vector for the noise can lead to instability or overfitting.

•	The loss function requires a specific penalty term to prevent the generated noise from becoming infinitesimally small.

•	There is a need to make architectural decisions regarding the generator's complexity and the type of noise distribution to use.

---

> ### Author Rebuttal · Authors · 2026-03-28
>
> We sincerely thank the reviewer for the positive comments and insightful questions. Some suggestions do **provide us further inspiration** and we will continue the exploration! Here is our response:
>
> ### Reply to Q1
>
> Yes, different auxiliary distributions may change the proof. The choice of Gaussian distribution is from a mathematical observation: The entropy calculation contains the $\log(\cdot)$ operator, which indicates that **the distributions from the exponential family (e.g., Gaussian, Laplacian) *may* eliminate the $\log$**. The exponential family is usually defined as
> $$
> f(x|\theta) = h(x) \exp(\theta^T \cdot T(x) - A(\theta))
> $$
> where $\theta$ is the "hyper-parameter" of the distribution. Clearly, only when $h(x)$ is constant, the $\log()$ in the entropy can vanish. From Eq. (18) in Appendix, we can find that the brief final formulation profits from **both choosing a Gaussian distribution and putting the loss to the covariance**. It may be not unique, but it may be the simplest scheme.
>
> ### Reply to Q2
>
> It depends on how much data the $\pi$-noise generator has seen during the training of contrastive model. If a noise generator is trained with large-scale pretraining representation models (e.g., DINOv2/v3), it may generate effective augmentations for most visual data, since it has probably seen similar images.
>
> On the dataset that the generator has never seen, it probably fails to generate the noisy augmentation, since **the noise generator of PiNDA is designed for improving its contrastive model**. In the failure case, **its contrastive model will also fail to produce the high-quality representations**.
>
> ### Reply to Q3
>
> The weight of the penalty for all experiments is **fixed as 1**. The performance of PiNDA is not sensitive to hyperparameters.
>
> Since we prefer to learn the Gaussian $\pi$-noise with $\mu=0$, the magnitude only impacts the learnable covariance matrix and therefore the magnitude is not important, as long as it is non-zero or not too small.
>
> ### Reply to Q4
>
> It is an **interesting and insightful suggestion**! In the current design, the generator and base contrastive model share the same architecture but no parameters. We may reduce the additional costs by sharing the weights, like siamese networks. We will try to extend our work by incorporating your suggestions.
>
> ### Reply to Q5
>
> If $\mu$ is learned during training, the learned noisy augmentations are more like perturbations,  similar to adversarial augmentations (*while the optimization objective is reverse*). As shown in the experiments on non-vision datasets, the adversarial method (CLAE) is quite instable (PiNDA with $\mu\neq0$, however, still outperforms it). More interestingly, when $\mu=0$, the learnable distribution is more like a "noise" instead of a "augmentation".
>
> In addition, we **visualize the learned $\mu$ in Table 4** of the main paper (P5). It shows that the learned $\mu$ contains no contour information. A possible reason may be the too many parameters, which increase the difficulty of training generator.
>
> ### Reply to Q6
>
> In fact, the augmented data point, $x'= x+\varepsilon$, is just **used for training** the contrastive model. So even when the feature is discrete and $x'$ is invalid, it will **not affect the regular inference of a real data point**. That's the main advantage of PiNDA on non-vision data.
>
> ### Reply to Q7
>
> No, the noise learned with different contrastive backbones is quite different, so the improvement of PiNDA differs a lot.

---

> > ### Author Rebuttal · Reviewer_gLQb · 2026-04-05
> >
> > Primary concern mentioned in the weaknesses are not addressed.

---

> > > ### Author Response · Authors · 2026-04-05
> > >
> > > Dear Reviewer gLQb,
> > >
> > > Sorry for the inconvenience since we find that the questions is consistent with the weaknesses.  We want to reply to the weakness about the primary concern, additional overhead, which is also raised by Reviewer efu4.
> > >
> > > >   (**The additional overhead is worthwhile**, which is agreed by Reviewer efu4)
> > >
> > > We agree that PiNDA increases some additional computation burden. However, the additional cost depends on the architecture of the noise generator. The cost on non-vision data is low, due to the simple implementation (a network consisting of 3 fully connected layers) of the noise generator.
> > >
> > > We run some experiments on CIFAR to study the performance (by changing the batch size) **when the total costs are similar in Table 11 (Appendix, P16)**. We found that, even after increasing the batch size so that the base SimCLR consumes a similar GPU memory, **PiNDA (with a smaller batch size) still outperforms the original SimCLR**. It verifies that **the improvements are not simply due to more computational costs**. Here is Table 11:
> > >
> > > |                  | CIFAR-10   |        |                |                | CIFAR-100  |        |                |                |
> > > | :--------------- | :--------- | :----- | :------------- | :------------- | :--------- | :----- | :------------- | :------------- |
> > > |                  | Batch Size | Memory | kNN            | SR             | Batch Size | Memory | kNN            | SR             |
> > > | SimCLR           | 256        | 15.85G | 74.43±0.52     | 84.81±0.91     | 256        | 15.70G | 43.59±1.00     | 57.67±2.20     |
> > > | SimCLR           | 300        | 19.06G | 72.44±0.23     | 85.76±0.55     | 300        | 19.07G | 43.83±0.83     | 58.82±1.60     |
> > > | **PiNDA+SimCLR** | 256        | 18.11G | **75.00±0.12** | **86.48±0.97** | 256        | 18.10G | **45.09±0.99** | **59.97±1.33** |
> > >
> > > >   (**Further improvement in the future**)
> > >
> > > As suggested by Question-4, it is a quite promising scheme to **introduce the weight sharing** mechanism to the noise generator, since **the most powerful implementation of noise generator is usually a similar architecture of base model**. For example, it would be better to use ResNet18 as noise generator for SimCLR-ResNet50. Your suggestion is really inspirational! We will explore the weight sharing implementation in the future. Thanks again for your constructive comments.

---

### Official Review · Reviewer_efu4 · 2026-03-13

**Soundness:** 3
**Presentation:** 4
**Significance:** 3
**Originality:** 3
**Overall Recommendation:** 5
**Confidence:** 3

**Summary:**

This paper establishes the connection between contrastive learning and π-noise. Through this connection, the predefined (image) data augmentations in the standard contrastive learning can be viewed as a point estimation of π-noise. Moreover, this paper proposes a π-noise generator to automatically learn the π-noise as the augmentation, which applies to all data types beyond images. In experiments, the proposed method serves as an extra module for all contrastive models, and consistently improves the model performance.

**Compliance With Llm Reviewing Policy:**

Affirmed.

**Final Justification:**

The rebuttal solved my primary concern about the computational overhead on the vision data.

I also read other reviewers' comments. Regarding the insufficient performance gain on large vision datasets (raised by Reviewer uMqq who gives a negative score), I think we should not criticize too much on this point. For me, the primary contribution of this paper is to "simulate" vision data augmentations with $\pi$-noise, so as to possibly extend the vision data augmentations to general data types, so a comparable or slightly better performance (with comparable computational overhead) on the vision datasets is acceptable to me.

For the above reasons, I keep my original recommendation score and lean towards acceptance.

**Key Questions For Authors:**

1. How does PiNDA perform on non-vision data with more classes?

**Limitations:**

Yes.

**Strengths And Weaknesses:**

**Strength**
1. This paper is well written and very easy to follow.
2. The method is well-motivated by information theory.
3. The proposed noise-driven data augmentation can be applied to any contrastive models, and especially can be applied to non-image data.
4. The performance is validated by extensive experiments.

**Weakness**
1. The additional cost of PiNDA on image data is high (both time and space). The additional cost on non-image data is acceptable though.
2. The number of classes of the selected non-vision data is relatively small.

---

> ### Author Rebuttal · Authors · 2026-03-28
>
> We greatly appreciate your appreciation of our contributions. Regarding the weaknesses and questions, we prepare the response item by item.
>
> ### Reply to Weakness-1
>
> > The additional cost of PiNDA on image data is high (both time and space). The additional cost on non-image data is acceptable though.
>
> The additional cost depends on the architecture of the noise generator. The low cost on non-image data is due to the simple implementation (a network consisting of 3 fully connected layers) of the noise generator.
>
> We run some experiments on CIFAR to study the performance (by changing the batch size) **when the total costs are similar in Table 11 (Appendix, P16)**. We found that, even after increasing the batch size so that the base SimCLR consumes a similar GPU memory, **PiNDA (with a smaller batch size) still outperforms the original SimCLR**.  It verifies that **the improvements is not simply due to more computational costs**. Here is Table 11:
>
> |                  | CIFAR-10   |        |                |                | CIFAR-100  |        |                |                |
> | ---------------- | ---------- | ------ | -------------- | -------------- | ---------- | ------ | -------------- | -------------- |
> |                  | Batch Size | Memory | kNN            | SR             | Batch Size | Memory | kNN            | SR             |
> | SimCLR           | 256        | 15.85G | 74.43±0.52     | 84.81±0.91     | 256        | 15.70G | 43.59±1.00     | 57.67±2.20     |
> | SimCLR           | 300        | 19.06G | 72.44±0.23     | 85.76±0.55     | 300        | 19.07G | 43.83±0.83     | 58.82±1.60     |
> | **PiNDA+SimCLR** | 256        | 18.11G | **75.00±0.12** | **86.48±0.97** | 256        | 18.10G | **45.09±0.99** | **59.97±1.33** |
>
>
>
> ### Reply to Weakness-2
>
> > The number of classes of the selected non-vision data is relatively small.
>
> We added the experiments on Reuters-21578, containing 46 classes, which are reported as follows:
>
> |            | kNN            | SR             |
> | ---------- | -------------- | -------------- |
> | Random     | 38.30±0.47     | 44.21±0.14     |
> | SimCL      | 38.00±0.22     | 44.34±0.40     |
> | CLAE       | 38.95±0.39     | 41.91±0.10     |
> | CLAE+PiNDA | **39.18±0.26** | 42.07±0.50     |
> | PiNDA      | 38.40±0.06     | **44.37±0.27** |
>
> As shown above, **PiNDA still works more stably on the non-vision dataset with more classes**, compared with other heuristic methods.

---

> > ### Author Rebuttal · Reviewer_efu4 · 2026-04-04
> >
> > Thank you for the rebuttal. The additional experiments solve my concerns. I will keep my score.

---

> > > ### Author Response · Authors · 2026-04-04
> > >
> > > Dear Reviewer efu4,
> > >
> > > Many thanks for your supporting and acknowledgement! Your suggestions help us improve the quality of the paper. We will add the additional experiments to the paper in the final version.
> > >
> > > Best regards,
> > >
> > > Authors of Submission 2802

---

### Official Review · Reviewer_uMqq · 2026-03-15

**Soundness:** 3
**Presentation:** 2
**Significance:** 2
**Originality:** 3
**Overall Recommendation:** 4
**Confidence:** 3

**Summary:**

The paper studies the relationship between data augmentation in contrastive learning and the concept of positive-incentive noise (π-noise). The authors introduce an auxiliary Gaussian distribution derived from the contrastive loss in order to define a notion of task entropy for contrastive learning. Based on this formulation, they show that predefined data augmentations used in standard contrastive learning can be interpreted as a point estimation of π-noise.

Based on this connection, the paper proposes a framework called PiNDA that learns a noise generator to produce augmentations instead of relying on predefined transformations. The noise generator is trained jointly with the contrastive model using a reparameterization formulation. Experiments are conducted on several vision datasets and non-vision datasets to evaluate the proposed method when combined with existing contrastive learning models

**Compliance With Llm Reviewing Policy:**

Affirmed.

**Final Justification:**

The results on ImageNet (SimCLR + PiNDA vs SimCLR) have resolved my concerns. I am happy to increase my rating to weak accept.

**Key Questions For Authors:**

1. The task entropy definition relies on introducing an auxiliary Gaussian variable linked to the contrastive loss. Could the authors clarify whether this construction is theoretically unique or simply one of many possible choices?

2. How does PiNDA differ from prior methods that learn augmentations or perturbations (e.g., adversarial augmentation, latent-space augmentation)?

**Limitations:**

yes

**Strengths And Weaknesses:**

1. The paper provides an information-theoretic interpretation of data augmentation in contrastive learning, linking it to the π-noise framework. However, the connection between contrastive loss and the auxiliary Gaussian distribution appears somewhat heuristically constructed. The introduction of the auxiliary variable $\alpha$ and the definition of task entropy rely on design choices that are not strongly justified theoretically. This part needs more justification and explanation.

2. While the theoretical interpretation is novel, the method itself resembles existing learnable augmentation or perturbation approaches. The practical algorithm essentially introduces a learnable noise generator optimized jointly with the contrastive objective, which is conceptually similar to several prior methods.

3. Although improvements are reported across datasets, the magnitude of improvements is generally small (often ~0.5–2%) in many settings (both in non-vision and vision domains). Then the significance of introducing this conceptual framework and corresponding method is hard to justify as in learning the noise generator also introduces additional computational overhead.

---

> ### Author Rebuttal · Authors · 2026-03-29
>
> We sincerely thank you for the comments and suggestions, which **acknowledges our novelty**. We prepare the response item by item. **We are looking forward to your discussions**.
>
> ### Reply to W1&Q1
>
> It is indeed a deep question! The construction is **not unique**. As explained below, the Gaussian variable is a concise choice.
>
> We do admit that the definition of task entropy depends on design choices, since **it is actually an open problem**. Any random variable correctly reflecting the task complexity can be used to define the task entropy.
>
> > **(Why define task entropy using loss)**
>
> However, we aim at **a general framework** for different tasks, *not only limited to contrastive learning*. So **the definition should be built on an element which most ML models have**. The loss is therefore a perfect choice.
>
> > **(Why choose a Gaussian variable, not other distributions)**
>
> To 	formulate the task entropy with loss, we have to convert the loss to a random variable. The choice **originates from a mathematical observation**: The entropy calculation contains the $\log(\cdot)$, which indicates that **the distributions from the exponential family (e.g., Gaussian, Laplacian) *may* eliminate the $\log$**. The exponential family is usually defined as
> $$
> f(x|\theta) = h(x) \exp(\theta^T \cdot T(x) - A(\theta))
> $$
> where $\theta$ is the "hyper-parameter". Clearly, only when $h(x)$ is constant, the $\log$ in the entropy can vanish. Compared with other distributions (e.g., multivariate Laplacian), the Gaussian distribution is **more concise and simple**. As **$\alpha$ is only an auxiliary variable to convert the loss to probabilistic space**, we prefer a simple scheme instead of a general discussion of auxiliary distributions.
>
> > **(Advantages of Gaussian distribution)**
>
> **With the auxiliary Gaussian variable**, **the original contrastive loss is remained**. The only change is the sampling from  $\pi$-noise distribution as augmentations, rather than some fixed augmentations. **It allows us to focus on the distribution learning without changing the raw contrastive design** (architecture, loss, etc.).
>
> ### Reply to W2&Q2
>
> The differences with the adversarial augmentations are mainly from the two aspects:
>
> - (*Optimization*) **The optimization of adversarial augmentations is a min-max optimization problem**, since the adversarial perturbations is learned by **maximizing the contrastive loss**. **The optimization of PiNDA is a pure minimization optimization problem**,  since the $\pi$-noise augmentation is learned by **minimizing the contrastive loss**.
> - (*Learnable perturbation*) The adversarial method learns **a perturbation** for each sample, while PiNDA learns a **distribution** instead. Therefore, PiNDA will **sample a perturbation** from the learned distribution in each step. It indicates that PiNDA may be more powerful, since PiNDA can sample countless perturbations.
>
> The adversarial method, **CLAE (NeurIPS 2020), serves as the main baseline on non-vision data**. In addition, we supplement the comparisons of CLAE on vision datasets:
>
> ||CIFAR10||CIFAR100||STL10||
> |-|-|-|-|-|-|-|
> ||kNN|SR|kNN|SR|kNN|SR|
> |SimCLR| 86.05±0.21     | 91.98±0.71     | 37.56±0.80     | 70.43±0.27     | 61.69±0.67     | 62.54±0.95     |
> |PiNDA| **86.60±0.39** |**92.25±0.07**|37.62±0.64|**70.64±0.44**|**62.31±0.25**|**63.68±0.19**|
> |CLAE|79.57±0.33|79.96±0.16|39.31±0.25|40.53±0.40| 48.72±0.69     | 54.23±1.44     |
> |CLAE+PiNDA|80.09±0.40|80.29±0.22|**39.62±0.15**|40.73±0.48| 49.79±0.12     |55.35±0.73|
>
> In particular, PiNDA is completely compatible with adversarial methods. From the experiments with CLAE (the above table and Table 2), we find:
>
> - Compared with "*Random Noise*", CLAE does not work on each dataset, while **PiNDA outperforms Random Noise on all datasets**.
> - "CLAE+PiNDA" always outperforms CLAE, which shows the great compatibility of PiNDA.
>
> On the other hand, SimCL, serving as a baseline on non-vision and vision data, is a latent-space augmentation method. We can find that SimCL even works worse than Random Noise, which indicates that **the augmentation in latent space may cause more unstable training of contrastive learning**.
>
> ### Reply to W3
>
> It should be emphasized that **the contrastive paradigm can be stably applied to non-vision data** due to PiNDA, which is **appreciated by the other two reviewers**.  As shown in Table 2, **PiNDA achieves almost/over 10% improvements** on HAR, Reuters, and MSLR-WEB30K (**column "SR"**).
>
> The experiments on vision data are used to validate the generalization and compatibility of PiNDA, under different contrastive backbones, dataset scales, and training settings.
>
> In addition, Table 11 in Appendix shows that, even after increasing the batch size so that the raw SimCLR consumes a similar GPU memory, **PiNDA (with a smaller batch size) still outperforms the original SimCLR**. It verifies that **the improvements are not simply due to more computational costs**.

---

> > ### Author Rebuttal · Reviewer_uMqq · 2026-04-04
> >
> > Thanks for the response. It partially addresses my concerns. One major remaining concern from my side is that the performance gain appears to be very limited on larger-scale datasets, such as ImageNet.

---

> > > ### Author Response · Authors · 2026-04-04
> > >
> > > Dear Reviewer uMqq,
> > >
> > > Thanks for your acknowledgement. We have to point out that there are two large-scale datasets consisting of over 1M data points, MSLR-WEB30K (containing **10,558,186 samples**) and ImageNet (containing **1,331,167 samples**).
> > >
> > > On MSLR-WEB30K, the improvement is **over 10%** compared with the baseline (Random Noise). It indicates that PiNDA is **a stable and powerful scheme for non-vision data**, where the extension to non-vision data (not limited to vision data) is **our core motivation**. It should be pointed out that the **existing** powerful and well-known **contrastive models** (e.g., SimCLR, BYOL, MoCo, SimSiam) **cannot be applied to MSLR-WEB30K** due to the limitation of augmentations.
> > >
> > > |                      | kNN            | SR             |
> > > | -------------------- | -------------- | -------------- |
> > > | Random Noise         | 57.88±1.49     | 33.78±6.31     |
> > > | SimCL                | 64.21±0.42     | 47.13±0.78     |
> > > | PiNDA ($\mu=0$)      | **69.62±0.02** | 49.55±0.11     |
> > > | PiNDA ($\mu \neq 0$) | 69.60±0.06     | **50.95±1.43** |
> > >
> > > On ImageNet, it should be emphasized that PiNDA **also achieves more than 3% (kNN) / 2% (SR) improvement**  on SimCLR-ResNet18, where SimCLR obtains the best results compared with other contrastive models based on ResNet18.
> > >
> > > | Model            | kNN                        | SR                         |
> > > | ---------------- | -------------------------- | -------------------------- |
> > > | SimCLR           | 52.05                      | 64.18                      |
> > > | **SimCLR+PiNDA** | **55.48** (3.43$\uparrow$) | **66.86** (2.68$\uparrow$) |
> > >
> > > As clarified above, our core contribution is to **extend the contrastive paradigm to diverse kinds of data**, not merely vision data. As the preceding wonderful works of contrastive models focus on the vision data, we therefore conduct **extensive experiments on vision data to verify the generalization and compatibility** of PiNDA. In our opinion, the performance gain should be evaluated on both non-vision and vision data, which is also supported by the other two reviewers.
> > >
> > > Could you please reconsider the recommendation? If you have any concerns or questions, please feel free to inform us and we will try our best to solve them.
> > >
> > > Best regards,
> > >
> > > Authors of Submission 2802

---

### Decision · Program_Chairs · 2026-04-30

**Decision:**

Accept (regular)

**Comment:**

The paper establishes a theoretical connection between contrastive learning and π‑noise, and proposes a learnable noise‑driven augmentation framework (PiNDA) that generalizes to non‑vision data. Three reviewers gave scores of Weak Reject, Accept, and Weak Accept.

Main concerns are (1) modest gains on large vision datasets (e.g., ImageNet), (2) additional computational overhead, and (3) justification of the Gaussian auxiliary distribution. The rebuttal partially alleviates these concerns by showing strong gains on large‑scale non‑vision data (MSLR‑WEB30K, >10% improvement), controlling for memory usage, and clarifying the design choice of Gaussianity.

While the empirical improvement on vision benchmarks is not dramatic, the theoretical insight and the ability to extend contrastive learning to non‑vision data are valuable. The paper is technically sound but the remaining limitations should be carefully addressed in the camera‑ready version.